

# HER: an information theoretic alternative for geostatistics

Stephanie Thiesen[1], Diego M. Vieira[2,3], Mirko Mälicke[1], J. Florian Wellmann[4], Uwe Ehret[1]

[1]Institute of Water Resources and River Basin Management, Karlsruhe Institute of Technology, Karlsruhe, Germany
[2]Department for Microsystems Engineering, University of Freiburg, Freiburg, Germany
[3]Bernstein Center Freiburg, University of Freiburg, Freiburg, Germany
[4]Computational Geosciences and Reservoir Engineering, RWTH Aachen University, Aachen, Germany

*Correspondence to*: Stephanie Thiesen (stephanie.thiesen@kit.edu)

**Abstract.** Interpolation of spatial data has been regarded in many different forms, varying from deterministic to stochastic,
purely data-driven to geostatistical, and parametric to non-parametric methods. In this study, we propose a stochastic,
geostatistical estimator which combines information theory with probability aggregation methods for minimizing predictive
uncertainty, and predicting distributions directly based on empirical probability. Histogram via entropy reduction (HER)
relaxes parametrizations, avoiding the risk of adding information not present in data (or losing available information). It
provides a proper framework for uncertainty estimation that takes into account both spatial configuration and data values,
while allowing to infer (or introduce) physical properties (continuous or discontinuous characteristics) of the field. We
investigate the framework utility using synthetically generated datasets and demonstrate its efficacy in ascertaining the
underlying field with varying sample densities and data properties (different spatial correlation distances and addition of noise).
HER shows comparable performance with popular benchmark models and the additional advantage of higher generality. The
novel method brings a new perspective of spatial interpolation and uncertainty analysis to geostatistics and statistical learning,
using the lens of information theory.

## 1 Introduction

Spatial interpolation methods are useful tools for filling gaps in data. Since information of natural phenomena is often collected
by point sampling, interpolation techniques are essential and required for obtaining spatially continuous data over the region
of interest (Li and Heap, 2014). There is a broad range of methods available that have been considered in many different forms,
from simple approaches such as nearest neighbors (NN) and inverse distance weighting (IDW) to geostatistical and, more
recently, machine learning methods.

Geostatistical, stochastic approaches, such as ordinary kriging (OK), have been widely studied and applied in various
disciplines since their introduction to geology and mining by Krige (1951), bringing powerful results in the environmental
science context. However, 'kriging', like other parametric regression methods (Yakowitz and Szidarovszky, 1985), relies on
prior assumptions about theoretical functions, and, therefore includes the risk of sub-optimal performance due to sub-optimal
user choices. If, on the one hand, OK is probabilistic and therefore offers as a result the estimator's uncertainty (through





variance), on the other hand, deterministic estimators (NN and IDW) avoid these function parametrizations at the cost of neglecting uncertainty analysis. In this sense, researchers are confronted with the trade-off between avoiding parametrization assumptions and obtaining uncertainty results (stochastic predictions).

More recently, with the increasing availability of data volume and computer power (Bell et al., 2009), machine learning (here referred to data-driven) methods have become increasingly popular as a substitute or complement to established modeling approaches. However, most of the popular data-driven methods have been developed in the computational intelligence community and since they are not built for solving particular problems, such as spatial interpolation, apply these methods remains a challenge for the researchers outside the computational intelligence (Solomatine and Ostfeld, 2008). And, even

though they enable automated learning from data in stochastic and non-parametric framework; according to Solomatine and Ostfeld (2008), they are not always appreciated for working essentially data-based, replacing the 'knowledge-driven' models describing physical behavior.

In the context of data-based modeling in environmental science, concepts and measures from information theory are being used for describing and inferring relations among data (Liu et al., 2016; Thiesen et al., 2019), quantifying uncertainty and

evaluating model performance (Chapman, 1986; Liu et al., 2016; Thiesen et al., 2019), estimating information flows (Weijs, 2011; Darscheid, 2017), and measuring similarity, quantity and quality of information in hydrological models (Nearing and Gupta, 2017; Loritz et al. 2018; Loritz et al. 2019). In the spatial context, information-theoretic measures were used to obtain the longitudinal profiles of rivers (Leopold and Langbein,1962), to derive rank-size rule for human settlements (Berry and Garrison, 1958; Curry, 1964), to explore the amount of information in spatial probability distributions for geographical

differentiations (Gurevich, 1969), to solve problems of spatial aggregation, analyze spatial redundancy and information gain and loss (Batty, 1974; Singh, 2013), to analyze spatiotemporal variability (Mishra et al., 2009; Brunsell, 2010), to measure spatial dissimilarity (Naimi, 2015), similarity and complexity (Pham, 2010), to analyze spatial uncertainty (Wellmann, 2013), to assess the risk of landslides (Roodposhti et al., 2016), and to describe spatial heterogeneity (Bianchi and Pedretti, 2018).

According to Solomatine and Ostfeld (2008), the main challenges for researches in hydroinformatics to apply data-driven

methods lie in testing various combinations of methods for particular water-related problems, in combining them with optimization techniques, in developing robust modelling procedures able to work with noisy data, and in developing methods providing the adequate model uncertainty estimates. To overcome these challenges in the framework of spatial interpolation and the mentioned tradeoff (parametrization and uncertainty), this paper is concerned with formulating and testing a novel method based on principles of geostatistics, data-based modeling, information theory and probability aggregation methods to

describe spatial patterns and solve spatial interpolation problems. In order to avoid fitting spatial correlation functions and making assumptions about the underlying distribution of the residues or about uncertainty, it relies on empirical, discrete probability distributions to: extract the spatial dependence structure of the field, minimize entropy of predictions, and produce a probabilistic interpolation. Thus, the proposed histogram via entropy reduction (HER) approach allows non-parametric and stochastic predictions, avoiding the shortcomings of fitting deterministic curves and therefore the risk of adding information

that is not contained in the data (or losing available information), but still relying on geostatistical concepts. HER is seen as an





in-between geostatistics (knowledge-driven) and statistical learning (data-driven) in the sense that it allows automated learning from data, bounded in a geostatistical framework.

Our experimental results show that the proposed method is flexible for combining distributions in different ways and presents comparable performance to ordinary kriging for various sample sizes and field properties (short and long range, with and

without noise). Furthermore, we show that its potential goes beyond data prediction, since, by construction, HER allows inferring (or introducing) physical properties (continuity or discontinuity characteristics) of a field under study, and provides a proper framework for uncertainty prediction, which takes into account not only the spatial configuration of the data, as is the case for geostatistical procedures like kriging (Bárdossy and Li, 2008), but also the data values.

The paper is organized as follows. The method is presented in Sect. 2. In Sect. 3, we describe the data properties, performance

parameters, validation design and benchmark models. In Sect. 4 we explore the properties of three different aggregation methods, present the results of HER for different samples sizes and data types and compare them to benchmark models, and, in the end, discuss the achieved outcomes and model contributions. Finally, we draw conclusions in Sect. 5.

## 2 Method description

The core of HER has three main steps: i) characterization of spatial correlation; ii) selection of optimal weights via entropy

minimization; and iii) prediction of the target probabilities (which uses the spatial structure and optimal weights to interpolate distributions for unsampled targets). The first and third steps are shown in Fig. 1.

In the following sections, we start with a brief introduction of information theory measures employed in the method, and then describe in detail all the three method steps.

### 2.1 Information theory

Information theory provides a framework for measuring information and quantifying uncertainty. In order to extract the spatial correlation structure from observations and to minimize the uncertainties of predictions, two information theoretic measures are used in HER and will be described here. This section is based on Cover and Thomas (2006), which we suggest for a more detailed introduction to concepts of information theory.

The entropy of a probability distribution can be seen as a measure of the average uncertainty in a random variable. The measure,

first derived by Shannon (1948), varies from zero to infinity and it is additive for independent events (Batty, 1974). The formula of Shannon entropy ($H$) for a discrete random variable $X$ with a probability $p(x)$, $x \in \chi$, is defined by

$$H(X) = - \sum_{x \in \chi} p(x) \log_2 p(x).$$

(1)

We use the logarithm to base two, so that the entropy is expressed in unit bits. Each bit corresponds to an answer to one optimal yes/no question asked with the intention of reconstructing the data. In the study, Shannon entropy is used to extract the infogram and range (correlation length) of the dataset (explored in more depth in Sect. 2.2).





Besides quantifying the uncertainty of a distribution, it is also possible to compare similarities of two probability distributions
$p$ and $q$ using the Kullback-Leibler divergence ($D_{KL}$). Comparable to the expected logarithm of the likelihood ratio (Cover and
Thomas, 2006; Allard et al., 2012), the Kullback-Leibler divergence quantifies the statistical 'distance' between two probability
mass functions $p$ and $q$ using the following equation

$$D_{KL}(p||q) = \sum_{x \in \chi} p(x) \log_2 \frac{p(x)}{q(x)}. \tag{2}$$

Also referred to as relative entropy, $D_{KL}$ can be understood as a measure of information loss of assuming that the distribution
is $q$ when in reality it is $p$ (Weijs et al., 2010). It is always nonnegative and is zero strictly if $p = q$. In the HER context,
Kullback-Leibler divergence is used to select the best weights for aggregating distributions (detailed in Sect. 2.3). The measure
was also used as a scoring rule for performance verification of probabilistic predictions (Gneiting and Raftery, 2007, and Weijs
et al., 2010).

Note that the measures presented by Eqs. (1) and (2) are defined as functionals of probability distributions, not depending on
the variable $X$ value or its unit. This is favorable, as it allows joint treatment of many different sources and sorts of data in a
single framework.

## 2.2  Spatial characterization

The spatial characterization (Fig. 1a) is the first step of HER. It consists of quantifying the information available in data and
of using it to infer its spatial correlation structure. For capturing the spatial variability and related uncertainties, concepts of
geostatistics and information theory are incorporated into the method. As shown in Fig. 1a, the spatial characterization phase
aims, through the infogram cloud, to obtain: $\Delta z$ probability mass functions (PMFs), where $z$ is the variable under study; the
infogram; and, finally, the correlation length (range). These outputs are outlined in Fig. 2 and attained in the following steps:

   i. (Fig. 2a): Calculate the difference of the $z$-values ($\Delta z$) between pairs of observations; associate each $\Delta z$ to the
      Euclidean separation distance of its respective point pairs. Define the lag distance (demarcated by red dashed lines),
115       here called distance classes, or simply classes. Divide the range of $\Delta z$ values into a set of bins (demarcated by
      horizontal gray lines).

   ii. (Fig. 2b): For each distance class, construct the $\Delta z$ PMF from the $\Delta z$ values inside the class (conditional PMFs). Also
       construct the $\Delta z$ PMF from all data in the dataset (unconditional PMF).

   iii. (Fig. 2c): Calculate the entropy of each $\Delta z$ PMF; calculate the entropy of the unconditional PMF. Compute the range
120        of the data: this is the lag class where the conditional entropy exceeds the unconditional entropy. Beyond this point,
       the neighbors start becoming un-informative, and it would be pointless to use information outside this neighborhood.

The infogram plays a role similar to that of the variogram: Through the lens of information theory, we can characterize the
spatial dependence of the dataset, calculate the spatial (dis)similarities, and compute its correlation length (range). It describes
the statistical dispersion structure of pairs of observations for the distance class separating these observations. Quantitatively,





it is a way of measuring the uncertainty about Δz given the class. Graphically, the infogram shape is the fingerprint of spatial dependence, where the larger the entropy of one class, the more uncertain (disperse) its distribution is. It reaches a threshold, where the data no longer show significant spatial correlation. This procedure, besides guaranteeing less uncertainty in the results (since we are using the more informative relations through the classes), reduces the number of classes to be mapped, thus improving the speed of calculation.

Naimi (2015) introduced a similar concept to the infogram called entrogram, which is used for the quantification of the spatial association of both continuous and categorical variables. In the same direction, Bianchi and Pedretti (2018) employed the term entrogram for quantifying the degree of spatial order and ranking different structures. Both works, as well as the present study, are carried out with variogram-like shape, entropy-based measures, and looking for data (dis)similarity, yet with different purposes and metrics. The proposed infogram terminology seeks to provide an easy-to-follow association with the

quantification of information available in the data.

The spatial characterization stage provides a way of inferring conditional distributions of the target given its observed neighbors without the need, for example, of fitting a theoretical correlation function. The way we can combine the distributions and the contribution weight of each neighbor are topics of the next section.

Converting the frequency distributions of Δz into probability mass function (PMF) requires a cautious choice of bin width,

since this decision will frame the PMFs which will be used as a model and directly influence the statistics we compute for evaluation ($D_{KL}$). Many methods for choosing an appropriate binning strategy have been suggested (Knuth, 2013; Gong et al., 2014; Pechlivanidis et al., 2016; Thiesen et al., 2018). Throughout this paper, we will stick to equidistant bins, since they have the advantage of being simple, computationally efficient (Ruddell and Kumar, 2009), and introduce minimal prior information (Knuth, 2013). The bin size was defined based on Thiesen et al. (2018), by comparing the cross entropy ($H_{pq} = H(p) +$

$D_{KL}(p||q)$) between the full learning set and subsamples for various bin widths. Furthermore, as recommended by Darscheid et al. (2018), we assigned a small probability equivalent to the probability of a single-pair-point count to all bins in the histogram after converting it to a PMF by normalization, to assure nonzero probabilities when estimating distributions.

## 2.3 Minimization of estimation entropy

For inferring the conditional distribution of the target (unknown point) $z_0$ given each one of the known $z_i$ observations (where

$i = 1, ..., n$ are the indices of the sampled observations), we used the Δz PMFs obtained at the spatial characterization step (Sect. 2.2). To do so, each neighbor $z_i$ (known observation) is associated to a class, and hence to a Δz distribution, according to their distance to the target $z_0$. This implies the assumption that the empirical Δz PMFs apply everywhere in the field, irrespective of specific location, and only depend on the distance between points (distance class). Each Δz PMF is then shifted by the $z_i$ value of the observation it is associated with, yielding the z PMF of the target given the neighbor $i$. Assume for





instance three observations $D_1, D_2, D_3$ from the field and that we want to predict the probability distribution of the target $A$. In this case, what we infer at this stage are the conditional probability distributions $P(A|D_1)$, $P(A|D_2)$, and $P(A|D_3)$.

Now, since we are in fact interested in the probability distribution of the target conditioned to multiple observations $P(A|D_1, D_2, D_3)$, how can we optimally combine the information gained from individual observations to predict this target probability? In the next sections, we address this issue by using aggregation methods. After introducing potential ways to

combine PMFs (Sect. 2.3.1), we propose an optimization problem via entropy minimization for defining the weight parameters needed for the aggregation (Sect. 2.3.2).

### 2.3.1  Combining distributions

The problem of combining multiple conditional probability distributions into a single one is treated here by using aggregation methods. This subsection is based on the work by Allard et al. (2012), which we recommend as a summary of existing

aggregation methods (also called opinion pools), with a focus on their mathematical properties.

The main objective of this process is to aggregate probability distributions $P_i$ coming from different sources into a global probability distribution $P_G$. For this purpose, the computation of the full conditional probability $P(A|D_1, \ldots, D_n)$ – where $A$ is the event we are interested in (the target) and $D_i$, $i = 1, \ldots, n$ is a set of data events (or neighbors) – is done by the use of an aggregation operator $P_G$, called pooling operator, such that

$$P(A|D_1, \ldots, D_n) \approx P_G\big(P(A|D_1), \ldots, P(A|D_n)\big). \tag{3}$$

From now on, we will adopt a similar notation to that of Allard et al. (2012), using the more concise expressions $P_i(A)$ to denote $P(A|D_i)$ and $P_G(A)$ for $P_G\big(P_1(A), \ldots, P_n(A)\big)$.

The most intuitive way of aggregating the probabilities $P_1, \ldots, P_n$ is by linear pooling, which is defined as

$$P_{G_{OR}}(A) = \sum_{i=1}^{n} w_{OR_i} P_i(A), \tag{4}$$

where $n$ is the number of neighbors, and $w_{OR_i}$ are positive weights verifying $\sum_{i=1}^{n} w_{OR_i} = 1$. Eq. (4) describes mixture models in which each probability $P_i$ represents a different population. If we set equal weights $w_{OR_i}$ to every probability $P_i$, the method

reduces to an arithmetic average, coinciding with the disjunction of probabilities proposed by Tarantola and Valette (1982) and Tarantola (2005). Since it is a way of averaging distributions, the resulting probability distribution $P_{G_{OR}}$ is often multi-modal. Additive methods, such as linear pooling, are related to union of events and to the logical operator OR.

Multiplication of probabilities, in turn, is described by the logical operator AND, and it is associated to the intersection of events. One aggregation method based on the multiplication of probabilities is the log-linear pooling operator, which is defined

by





$$\ln P_{G_{AND}}(A) = \ln Z + \sum_{i=1}^{n} w_{AND_i} \ln P_i(A),\tag{5}$$

or equivalently

$$P_{G_{AND}}(A) \propto \prod_{i=1}^{n} P_i(A)^{w_{AND_i}},\tag{6}$$

where $Z$ is a normalizing constant, $n$ is the number of neighbors, and $w_{AND_i}$ are positive weights. One particular case consists

of setting $w_{AND_i} = 1$ for every $i$. This refers to the conjunction of probabilities proposed by Tarantola and Valette (1982) and

Tarantola (2005). In contrast to linear pooling, log-linear pooling is typically unimodal and less dispersed.

The aggregation methods are not limited to log-linear and linear pooling presented here. However, the selection of these two

different approaches to PMF aggregation seeks to embrace distinct physical characteristics of the field. The authors naturally

associate the intersection of distributions (AND combination, Eq. (5)) to fields with continuous properties. This idea is

supported by Journel (2002) when remarking that a logarithmic expression evokes the simple kriging expression (used for

continuous variables). For example, if we have two points $D_1$ and $D_2$ with different values and want to estimate the target

point $A$ in a continuous field, we would expect that the estimate at point $A$ would be somewhere between $D_1$ and $D_2$, which

can be achieved by an AND combination. In a more intuitive way, if we notice that, for kriging, the shape of the predicted

distribution is assumed to be fixed (Gaussian, for example), multiplying two distributions with the same variance and different

means would result in a Gaussian distribution too, less dispersed than the original ones, as also seen for the log-linear pooling.

On the other hand, Krishnan (2008) pointed out that the linear combination, given by linear pooling, identifies a dual indicator

kriging estimator (kriging used for categorical variables), which we see as an appropriate method for fields with discontinuous

properties. In this case, if we have two points $D_1$ and $D_2$ belonging to different categories, target $A$ will either belong to the

category of $D_1$ or $D_2$, which can be achieved by an OR combination. In other words, the OR aggregation is a way of combining

information from different sides of the truth, thus, a conservative way of considering the available information from all sources.

Note that, for both linear and log-linear pooling, weights equal to zero will lead to uniform distributions, therefore bypassing

the PMF in question. Conveniently, the uniform distribution is the maximum entropy distribution among all discrete

distributions with the same finite support.

The selection of the most suitable aggregation method depends on the specific problem (Allard et al., 2012), and it will

influence the PMF prediction and, therefore, the uncertainty structure of the field. Thus, depending on the knowledge about

the field, a user can either add information to the model by applying an a-priori chosen aggregation method or infer these

properties from the field. Since, in practice, there is often a lack of information to accurately describe the interactions between

the sources of information (Allard et al., 2012), inference is the approach we tested for the comparison analysis (Sect. 4.2).





For that, we propose to estimate the distribution of a target $P_G$ by combining $P_{G_{AND}}$ and $P_{G_{OR}}$ using the log-linear pooling operator, such that

$$P_G(A) \propto P_{G_{AND}}(A)^\alpha P_{G_{OR}}(A)^\beta, \tag{7}$$

where $\alpha$ and $\beta$ are positive weights varying from 0 to 1, which will be found by optimization. Eq. (7) was a choice made by
the authors as a way of balancing both natures of PMF aggregation. The idea is to find the appropriate proportion of $\alpha$
(continuous) and $\beta$ (discontinuous) properties of the field by minimizing relative estimation entropy. The equation is based on
the log-linear aggregation, as opposed to linear aggregation, since the latter is often multi-modal, which is an undesired
property for geoscience applications (Allard et al., 2012). Alternatively, Eqs. (4) or (5) or a linear combination of $P_{G_{AND}}(A)$

and $P_{G_{OR}}(A)$ could be used. We explore the properties of the pure linear and log-linear pooling in Sect. 4.1.
The following section addresses the optimization problem for estimating the weights of the aggregation methods of Eqs. (4),
(5) and (7).

### 2.3.2 Weighting PMFs

Scoring rules assess the quality of probabilistic estimations (Gneiting and Raftery, 2007) and can be used for estimating the
parameters of a pooling operator (Allard et al., 2012). The logarithmic score proposed by Good (1952) and reintroduced from
an information-theoretical point of view by Roulston and Smith (2002) are strictly proper scoring rules (Gneiting and Raftery,
2007) since they provide summary metrics of performance that address calibration and sharpness simultaneously, by rewarding
narrow prediction intervals and penalizing intervals missed by the observation (Gneiting and Raftery, 2007). According to
Gneiting and Raftery (2007), the divergence function associated with the logarithmic score is the Kullback-Leibler divergence
($D_{KL}$, Eq. (2)), which we used for selecting the proportion of the log-linear and linear pooling ($\alpha$ and $\beta$, Eq. (7)), as well as the
$w_{OR}$ and $w_{AND}$ weights (Eq. (4) and (5), respectively), here generalized as $w$.

By means of leave-one-out cross-validation (LOOCV), the optimization problem is then defined in order to find the set of $w$
which minimizes the expected relative entropy ($D_{KL}$) of all targets. The idea is to choose weights such that the disagreement
of the 'true' distribution (or observation value, when no distribution is available) and estimated distribution is minimized. In
Eq. (4) and (5) we assign one weight for each distance class $k$. This means that, given a target $z_0$, the neighbors grouped in the
same distance class will be assigned the same weight. For a more continuous weighting of the neighbors, as an extra step, we
linearly interpolate the weights according to the Euclidean distance and the weight of the next class. Another option could be
narrowing down the class width, in which case more data is needed to estimate the respective PMFs.

Firstly, we obtained in parallel the weights of Eqs. (4) and (5) by convex optimization, and later $\alpha$ and $\beta$ by grid search with
both weight values ranging from 0 to 1 (steps of 0.05 were used in the application case). In order to make the convex
optimization more well behaved, the following constraints were employed: i) for linear pooling, set $w_1 = 1$, to avoid non-

unique solutions; ii) force weights to decrease monotonically (i.e., $w_{k+1} \leq w_k$); iii) define a lower bound, to avoid numerical instabilities (e.g., $w_k \geq 10^{-6}$); iv) define an upper bound ($w_k \leq 1$). Finally, after the optimization, normalize the weights to verify $\sum_{i=1}^{k} w_{OR_i} = 1$ for the linear pooling (for log-linear pooling, the resulting PMFs are normalized).

In order to increase computational efficiency, and due to the minor contribution of neighbors in distance classes far away from
the target, the authors only used the twelve neighbors closest to the target when optimizing $\alpha$ and $\beta$ and when predicting the target. Note that this procedure is not applicable for the optimization step using Eqs. (4) and (5), since we are looking for one weight $w$ for each class, and therefore we can't risk neglecting classes whose weights we have an interest in. For the optimization phase discussed here, as well as for the prediction phase (next section topic), the limitation of number of neighbors together with the removal of classes beyond range are efficient means of reducing the computational effort involved in both
phases.

## 2.4 Prediction

With the results of the spatial characterization step (classes, $\Delta z$ PMFs, and range, as described in Sect. 2.2), the definition of the aggregation method and its parameters (Sects. 2.3.1 and 2.3.2, respectively), and the set of known observations, we have the model available for predicting distributions.

Thus, for estimating a specific unknown point (target), first, we calculate the Euclidean distance from the target to its neighbors (known observations). Based on this distance, we obtain the distance class of each neighbor, and associate to each its corresponding $\Delta z$ PMF. As mentioned in Sect. 2.2, neighbors beyond the range are associated with the $\Delta z$ PMF of the full dataset. For obtaining the $z$ PMF of target $z_0$ given each neighbor $z_i$, we simply shift the $\Delta z$ PMF of each neighbor by its $z_i$ value. Finally, by applying the defined aggregation method, we combine the individual $z$ PMFs of the target given each
neighbor to obtain the predicted PMF of the target. Fig. 1b presents a scheme of the main steps for PMF prediction of one target.

## 3 Testing HER

For the purpose of benchmarking, this section presents the data used for testing the method, establishes the performance metrics, and introduces the calibration and test design. Additionally, we briefly present the benchmark interpolators used for
the comparison analysis and some peculiarities of the calibration procedure.

### 3.1 Data properties

To test the proposed method in a controlled environment, four synthetic 2D spatial datasets with grid size 100x100 were generated from known Gaussian processes. A Gaussian Process is a stochastic method that is specified by its mean and a covariance function, or kernel (Rasmussen and Williams, 2006). The data points are determined by a given realization of a





prior, which is randomly generated from the chosen kernel function and associated parameters. We use rational quadratic

kernel (Pedregosa et al., 2011) as the covariance function, with two different correlation lengths, namely 6 and 18 units, to

produce two datasets with fundamentally different spatial dependence. For both, short- and long-range fields, a white noise

was introduced given by Gaussian distribution with mean 0 and standard deviation equal to 0.5. The implementation is taken

from the Python library scikit-learn (Pedregosa et al., 2011). The generated set comprises i) a short-range field without noise

(SR0), ii) a short-range field with noise (SR1), iii) a long-range field without noise (LR0), and iv) a long-range field with noise

(LR1). Figure 3 presents the field characteristics (parameters and image) and their summary statistics. For each field, the

summary statistics for the learning, validation, and test subsets are included in Supplement S1.

### 3.2 Performance criteria

For elucidating differences in the predictive power of the models, a quality assessment was carried out with three criteria:

mean absolute error ($E_{MA}$), and Nash–Sutcliffe efficiency ($E_{NS}$), for the deterministic cases, and the mean of the logarithmic

score rule, based on the Kullback-Leibler divergence ($D_{KL}$), for the probabilistic cases. $E_{MA}$ was selected because it gives the

same weight to all errors, while $E_{NS}$ penalizes variance as it gives more weight to errors with larger absolute values. $E_{NS}$ also

shows a normalized metric (limited to 1) which favors general comparison. All three metrics are shown in Eqs. (8), (9) and

(2), respectively. The validity of the model can be asserted when the mean error is close to zero, Nash–Sutcliffe efficiency is

close to one, and mean of Kullback-Leibler divergence is close to zero. The deterministic performance coefficients are defined

as

$$E_{MA} = \frac{1}{n} \sum_{i=1}^{n} |\hat{z}_i - z_i^{sim}|, \tag{8}$$

$$E_{NS} = 1 - \frac{\sum_{i=1}^{n}(\hat{z}_i - z_i^{sim})^2}{\sum_{i=1}^{n}(\hat{z}_i - \bar{z})^2}, \tag{9}$$

where $\hat{z}_i$ and $z_i^{sim}$ are, respectively, the observation and the prediction at $i$th location, $\bar{z}$ is the mean of the observations, and $n$

is the number of predicted observations.

For the applications in the study, we considered that there is no true distribution available for the observations in all field types.

Thus, the $D_{KL}$ scoring rule was calculated by comparing the filling of the single bin where the observed value is located, i.e.,

in Eq. (2), we set $p$ equal to one for the corresponding bin and compare it to the probability of the same bin in the predicted

distribution. This procedure is just applicable to probabilistic models, and it enables to measure how confident the model is in

predicting the correct observation. In order to calculate this metric for ordinary kriging, we converted the predicted PDFs

(probability density functions) to PMFs employing the same bins used for HER.





### 3.3 Calibration and test design


For the purpose of benchmarking and to investigate the effect of sample size, we applied holdout validation as follows: Firstly we randomly shuffled the data, and then divided it in three mutually exclusive sets: one to generate the learning subsets (containing up to 2000 data points), one for validation (containing 2000 data points), and another 2000 data points (20% of the full dataset) as test set. We calibrated the models with learning subsets of sizes between 200 and 2000 observations in the

increments of 200, 400, 600, 800, 1000, 1500, and 2000. We used the validation set for fine adjustments and plausibility checks. For avoiding multiple calibration runs, the resampling was designed in a way that the learning subsets increased in size by adding new data to the previous subset, i.e., the observations of small sample sizes were always contained in the larger sets. The validation and test datasets were fixed for all performance analyses, independently of the analyzed sample size, to facilitate model comparison. This procedure also avoided variability of results coming from multiple random draws, since, by

construction, we improved the learning with growing sample size, and we evaluated the results always in the same test set. The test set was kept unseen until the final application of the methods, as a 'lock box approach' (Chicco, 2017), and its results were used for evaluating the model performance presented in Sect. 4. The summary statistics for the learning, validation, and test subsets are presented in Supplement S1.

### 3.4 Benchmark interpolators

In addition to presenting a complete application of HER (Sect. 4.1), a comparison analysis among the best-known and used methods for spatial interpolation in the earth sciences (Myers, 1993; Li and Heap, 2011) was performed (Sect. 4.2). Covering deterministic, probabilistic and geostatistical methods, three interpolators were chosen for the comparison, namely nearest neighbors (NN), inverse distance weighting (IDW), and ordinary kriging (OK).

As in HER, all of these methods assume that the similarity of two point values decrease with increasing distance. Since NN

simply selects the value of the nearest sample to predict the values at an unsampled point without considering the values of the remaining observations, it was employed as a baseline comparison. IDW, in turn, linearly combines the set of sample points for predicting the target, inversely weighting the observations according with their distance to the target. The particular case where the exponent of the weighting function equals two is the most popular choice (Li and Heap, 2008), and it is known as the inverse distance squared (IDS), which is also applied here.

OK is more flexible than NN and IDW, since the weights are selected depending on how the correlation function varies with distance (Kitanidis, 1997, p.78). The spatial structure is extracted by the variogram, which is a mathematical description of the relationship between the variance of pairs of observations and the distance separating these observations (also known as lag). It is also described as the best linear unbiased estimator (BLUE) (Journel and Huijbregts, 1978, p.57), which aims at minimizing the error variance, and provides an indication of the uncertainty of the estimate. The authors suggest Kitanidis

(1997) and Goovaerts (1997) for a more detailed explanation of variogram and OK, and Li and Heap (2008) for NN and IDW.



NN and IDS do not require calibration. For calibrating HER aggregation weights, we applied LOOCV, as described in Section 2.3, optimizing the performance of the left-out sample in the learning set. As a loss function, minimization of $D_{KL}$ was applied. After learning the model, we used the validation set for a plausibility check of the calibrated model and eventually adjusting of parameters. Note that no curve fitting is needed for the application of HER.

For OK, the fitting of the model was applied in a semi-automated approach. The variogram range, sill and nugget were fitted to each of the samples taken from the four fields individually. They were selected by least squares (Branch et al., 1999). The remaining parameters, namely the semi-variance estimator, the theoretical variogram model, the minimum and the maximum number of neighbors considered during OK were jointly selected for each field type (SR and LR), since they derive from the same field characteristics. This means that for all sample sizes of SR0 and SR1 the same parameters were used, except range,
sill and nugget, which were fitted individually to each sample size. The same applies to LR0 and LR1. These parameters were chosen by expert decision, supported by comparison and cross-validation. Variogram fitting and kriging interpolation were applied using the scikit-gstat Python module (Mälicke and Schneider, 2019).

The selection of lag size has important effects on the infogram (HER) and, as discussed in Oliver and Webster (2014), on the empirical variogram (OK). However, since the goal of the benchmarking analysis was to find a fair way to compare the
methods, we fixed the lag distances of OK and distance classes of HER in equal intervals of two distance units (three times smaller than the kernel correlation length of the short-range dataset).

Since all methods are instance-based learning algorithms, due to the fact that the predictions are based on the sample of observations, the learning set is stored as part of the model and used in the test phase for the performance assessment.

## 4 Results and discussion

In this section, three analyses are presented. Firstly, we explore the results of three distinct models of HER using different aggregation methods on the synthetic dataset LR1 with learning set of 600 observations (Sect. 4.1). In Sect. 4.2, we summarize the results on synthetic datasets LR0, LR1, SR0, SR1 for all learning sets and compare HER performance with traditional interpolators. For brevity, the model outputs were omitted in the comparison analysis, and only the performances for each dataset and interpolator are shown. Finally, Sect. 4.3 discusses the probabilistic methods OK and HER, comparing their
performance and contrasting their different properties and assumptions. For all applications, the test set was used to assess the performance of the methods.

### 4.1 HER application

This section presents three variants of HER models applied to the LR1 field with a learning subset of 600 observations (LR1-600). This dataset was selected since, due to its optimized weight results $\alpha$ and $\beta$ (which reach almost the maximum value
(one) proposed in Eq. (7)), it favors to contrast uncertainty results of HER applying distinct aggregation methods. For LR1-600, the optimized weights are $\alpha = 1$ and $\beta = 0.95$.





As a first step, the spatial characterization of the selected field is obtained and shown in Figure 4. For brevity, only the odd classes are shown in Figure 4b. In the same figure, the Euclidean distance (in grid units) relative to where the class was extracted is indicated after the class in interval notation (left-open, right-closed interval). For both $z$ PMFs and $\Delta z$ PMFs, a bin width of 0.2 (10% of the distance class width) was selected and kept the same for all applications and performance calculations. Based on the infogram cloud (Figure 4a), the $\Delta z$ PMFs for all range classes were obtained. Then the range was identified as the class beyond which the entropy of the class PMFs exceeded the entropy of all data (class 23 corresponding to a Euclidean distance of 44 grid units), see the intersect of the blue line and the red dotted line in Figure 4b. In Figure 4c, it is also possible to notice a steep reduction in entropy (red curve) for furthest classes. It occurs due to the reduced number of pairs composing the $\Delta z$ PMFs. A similar behavior is also typically found in experimental variograms (not shown).

The number of pairs forming each $\Delta z$ PMFs, and the optimum weights obtained for Eqs. (4) and (5) are presented in Figure 5. Figure 5a shows the number of pairs which compose the $\Delta z$ PMFs by class, where the first class has just under 500 pairs and the last class inside the range (light blue) has almost 10,000 pairs. About 40% of the pairs (142,512 out of 359,400 pairs) are inside the range.

In Figure 5b, we obtained the weight of each class by convex optimization as described in Sect. 2.3.2 on the test data set. The dots in Figure 5b represent the optimized weights of each class. As expected, the weights reflect the decreasing spatial dependence of variable z with distance. Regardless of the aggregation method, the LR1-600 models are highly influenced by neighbors up to a distance of about 10 grid units (distance class 5).

For estimating $z$ PMFs of target points, three different methods were tested:

    i. Model 1: AND/OR combination, proposed by Eq. (7), where the optimized weights resulted in $\alpha = 1$ and $\beta = 0.95$;

    ii. Model 2: pure AND combination, given by Eq. (5);

    iii. Model 3: pure OR combination, given by Eq. (4).

The model results are summarized in Table 1 and illustrated in Figure 6, where the first column of the panel refers to the results of the AND/OR combination, the second column to the pure AND combination, and the third column to the pure OR combination. In Figure 6a and b, to assist in checking the heterogeneity (or homogeneity) of $z$ in the uncertainty maps (Figure 6b), the calibration set representation is scaled by its $z$ value, with the size of the cross increasing with $z$. In general, for the target identification, we use its grid coordinates (x,y).

Figure 6a shows the predicted $z$ mean (obtained from the predicted $z$ PMF) for the three analyzed models. Neither qualitatively (Figure 6a) nor quantitatively (Table 1) is it possible to differ the three models based on their mean or summary statistics of the predicted mean. Deterministic performance parameters ($E_{MA}$ and $E_{NS}$) are also quite similar among the tree models. However, in probabilistic terms, both the representation given by the entropy map (Figure 6b, which shows the entropy of the predicted $z$ PMFs) and the statistics of predicted $z$ PMFs together with the $D_{KL}$ performance (Table 1) reveal differences.

By construction, HER takes into account not only the spatial configuration of data but also the data values. In this fashion, targets close to known observations will not necessarily lead to reduced predictive uncertainty (or vice-versa). This is, e.g., the





case for targets A (10,42) and B (25,63). Target B (25,63) is located in between two sampled points in a heterogenous region (small and large z values, both in the first distance class), and presents distribution with bimodal shape and higher uncertainty (Figure 6c) especially for model 3 (4.68 bits). For the more assertive models (1 and 2), the distributions of target B (25,63) has lower uncertainty (Figure 6c, 3.42 and 3.52 bits, respectively). It shows some peaks, due to small bumps in the PMF neighbors (not shown) which are boosted by the $w_{AND}$ exponents in Eq. (5). In contrast, target A (10,42), which is located in a more

homogeneous region, with the closest neighbors in the second distance class, shows a sharper z PMF in comparison to target A (25,63) for models 1 and 3, and for all models a Gaussian-like shape.

Targets C (47,16) and D (49,73) are predictions for locations where observations are available. They were selected in regions with high and low z values to demonstrate the uncertainty prediction in locations coincident with the calibration set. For all three models, target C (47,16) presented lower entropy and $D_{KL}$ in comparison to target D (49,73), due to its distance to known

samples and homogeneity of z-values in the region.

Although the provided PMFs (Figure 6c) from models 1 and 2 present similar shapes, the uncertainty structure (color and shape displayed by the colors in Fig. 6b) of the overall field differs. Since model 1 is derived from the aggregation of models 2 and 3, as presented in Eq. (7), this combination is also reflected in its uncertainty structure, lying somewhere in-between model 2 and 3.

Model 1 is the bolder (more confident) model, since it has the smallest median entropy (3.45 bits). On the other hand, due to the averaging of PMFs, model 3 is the more conservative model, verified by the highest overall uncertainty (4.17 bits). Model 3 also predicted smaller minimum and higher maximum of mean values of z, as well, for the selected targets, and it provides the widest confidence interval.

The authors selected Model 1 (AND/OR combination) for the sample size and benchmarking investigation presented in the

next section.

## 4.2 Comparison analysis

In this section, HER was applied using the more confident AND/OR model proposed by Eq. (7). The test set was used to calculate the performance of all methods (NN, IDS, OK, and HER) as a function of sample size and dataset type (SR0, SR1, LR0, LR1). See Supplement S2 for the calibrated parameters of all models discussed in this section.

Figure 7 summarizes values of mean absolute error ($E_{MA}$), Nash–Sutcliffe efficiency ($E_{NS}$) and mean Kullback-Leibler divergence ($D_{KL}$) for all interpolation methods, sampling sizes, and dataset types. The SR field is presented in the left column and the LR in the right. Datasets without noise are represented by continuous lines and datasets with noise by dashed lines. $E_{MA}$ is presented in Figure 7a,b for the SR and LR fields, respectively. All models have the same order of magnitude of $E_{MA}$ for the noisy datasets (SR1 and LR1, dashed lines), with the performance of the NN model being the poorest, and OK being

slightly better than IDS and HER. For the datasets without noise (SR0 and LR0, continuous lines), OK performed better than the other models, with a decreasing difference given sample size. In terms of $E_{NS}$, all models have comparable results for LR

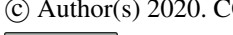



(Figure 7d), except NN in the LR1 field. A larger contrast in the model performances can be seen for the SR field (Figure 7c), where for SR1, NN performs worst and OK best. For SR0, especially for small sample sizes, OK performed better and NN poorly, while IDS and HER have similar results, with a slightly better performance for HER.

The probabilistic models OK and HER were comparable in terms of $D_{KL}$, with OK being slightly better than HER, especially for small sample sizes. An exception is made for OK in LR0. Since $D_{KL}$ scoring rule penalizes extremely confident but erroneous predictions, $D_{KL}$ of OK tended to infinity for LR0 and, therefore, it is not shown in Figure 7f.

For all models, the performance metrics for LR showed better results when compared to SR (compare left and right column in Fig. 7). The performance improvement given the sample size is similar for all models, as can be seen by the similar slopes of
the curves. In general, we noticed an improvement in the performance in SR fields up to a sample size of 1000 observations. On the other hand, in LR fields, learning process given sample sizes already stabilizes at around 400 observations. In addition to the model performance presented in this section, the summary statistics of the predictions and their residue correlation can be found in Supplement S3.

In this section we evaluated various models via direct comparison of performance measures. In the next section, we discuss
fundamental aspects of HER, with a focus on comparing it to ordinary kriging.

## 4.3 Discussion

Several important points emerge from this study. Because the primary objective was to explore the characteristics of HER, we first consider the effect of selecting the aggregation method (Sec. 4.1). Independent of the choice of aggregation method, the deterministic results (predicted mean of $z$) of all models were very similar. On the other hand, we could see different
uncertainty structures of the estimates for all three cases analyzed, ranging from a more confident method (AND/OR) to a more conservative one (OR). Considering that larger errors are expected in locations surrounded by data that are very different in value (Goovaerts, 1997, p.180, p.261), HER has proved effective in considering both spatial configuration of data and the data values regardless of the aggregation method selected.

As previously introduced in Sect. 2.3.1, the choice of aggregation method can happen beforehand in order to introduce physical
knowledge to the system, or several can be tested to learn about the response of the field to the selected model. Aside from their different mathematical properties, the motivation behind the selection of the two aggregation methods (linear and log-linear) was the incorporation of continuous or discontinuous field properties. The interpretation of the aggregation method is supported by Journel (2002) and Krishnan (2008) where the former connects a logarithmic expression to continuous variables and simple kriging), while the latter associates linear pooling to dual indicator kriging and therefore to categorical variables.
For example, if we have two points $D_1$ and $D_2$ with different values and want to estimate the target point $A$ in a continuous field, we would expect that the estimate at point $A$ would be somewhere between $D_1$ and $D_2$, which can be achieved by an AND combination. On the other hand, in the case of categorical data, and $D_1$ and $D_2$ belonging to different categories, target $A$ will either belong to the category of $D_1$ or $D_2$, which can be achieved by an OR combination.





As verified in Sec. 4.1, the OR (=averaging) combination of PMFs to estimate target PMFs was the most conservative (not
confident) method among all those tested. For this way of PMF merging, all distributions are considered feasible and each
point adds new possibilities to the result. On the other hand, AND combination of PMFs was a bolder approach, where we
intersect distributions to extract their agreements. In other words, we are narrowing down the range of possible values and the
final distribution satisfies all observations at the same time. Complementarily, considering the lack of information to accurately
describe the interactions between the sources of information, we proposed to infer $\alpha$ and $\beta$ weights (AND and OR
contributions, respectively) using Eq. (7). It turned out to be a good tradeoff between the pure AND and the pure OR model
and was hence used for benchmarking HER against traditional interpolation models.

In HER, the spatial dependence was analyzed by extracting $\Delta z$ PMFs and expressed by the infogram, where classes composed
by point pairs further apart were more uncertain (presented higher entropy) than classes formed by point pairs close to each
other. Aggregation weights (Supplement S2, Figure S2.1 and Figure S2.2) also characterize the spatial dependence structure
of the field. In general, as expected, noisy fields (SR1 and LR1) lead to smaller influence (weights) of the closer observations
than non-noisy datasets (Figure S2.1). In terms of $\alpha$ and $\beta$ contribution (Figure S2.2), while $\alpha$ received for all sample sizes
the maximum weight, $\beta$ increased with the sample size. As expected, in general, the noisy fields reflected a higher contribution
of $\beta$ due their discontinuity. For LR0, starting at 1000 observations, $\beta$ also stabilized at 0.55, indicating that the model
identified the characteristic $\beta$ of the population. The most noticeable result along these lines was that the aggregation method
directly influences the probabilistic results, and therefore the uncertainty (entropy) maps can be adapted according to the
characteristics of the variable or expert interest.

Although the primary objective of this study was to investigate the characteristics of HER, Sect. 4.2 compares it to three
traditional interpolation methods. In general, HER performed comparable to OK, the best method among the analyzed ones.
The probabilistic performance comparison was only possible between HER and OK, where both methods also produced
comparable results. Note that the datasets were generated using Gaussian Process, so that they remained within the
recommended settings for OK (field mean independent of location, normally distributed data and residues), thus favoring its
performance. Additionally, OK was also favored when converting their predicted PDFs to PMFs, since the defined bin width
was often orders of magnitude larger than the standard deviation estimated by OK. However, that was a necessary step for the
comparison, since HER does not fit PDFs for their predicted PMFs.

Especially for HER, which works with non-parametric PMFs, the number of distance classes and bin width basically defines
how accurate we want to be in the prediction. For comparison purposes, bin widths and distance classes were kept the same
for all models and were defined based on small sample sizes. However, with more data available, it would be possible to
describe better spatial dependence of the field in HER by increasing the number of distance classes and the number of PMF
bins. Although the increase in the number of classes would also affect OK performance (as it improves the theoretical
variogram fitting), it would allow more degrees of freedom for HER (since it optimizes weights for each distance class), which
would result in a more flexible model and closer reproducibility of data characteristics. In contrast, the degrees of freedom in





OK would be unchanged, since the number of parameters of the theoretical variogram does not depend on the number of classes.

HER does not require fitting of a theoretical function, its spatial dependence structure ($\Delta z$ PMFs, infogram) are derived directly
from the available data, while, according to Putter and Young (2001), OK predictions are only optimal if the weights are calculated from the correct underlying covariance structure, which in practice is not the case, since the covariance is unknown and estimated from the data. Thus, the choice of the theoretical variogram for OK can strongly influences the predicted $z$ depending on the data. In this sense, HER is more robust against user decisions compared to OK.

Considering the probabilistic models, both OK and HER present similarities. Both approaches take into consideration the
spatial structure of the variables, since their weights depend on the spatial correlation of the variable. Just as OK (Goovaerts, 1997, p.261), HER turned out to be a smoothing method, since the true values are overestimated in low-valued areas and underestimated in high-valued areas. However, as verified in Supplement S3 (Figure S3.1), HER revealed a reduced smoothing (residue correlation closer to zero) compared to OK for SR0, SR1 and LR1. In particular, for points beyond the range, both methods predict by averaging the available observations. While OK calculates the same weight for all observations beyond
the range and proceeds with their linear combination, HER associates $\Delta z$ PMF of the full dataset to all observations beyond the range and aggregates them using the same weight (weight of the last class).

OK and HER have different levels of generality: OK weights depend on how the fitted variogram varies in space (Kitanidis, 1997, p.78), HER weights take into consideration the spatial dependence structure of the data (via $\Delta z$ PMFs) and the $z$ values of the observations, since they are found by minimizing $D_{KL}$ between the true $z$ and its predicted distribution. In this sense, the
variance estimated by kriging ignores the observation values, retaining from the data only their spatial geometry (Goovaerts, 1997, p.180), while for HER, it is additionally influenced by the $z$ value of the observations.

Another important difference is that OK performs multiple local optimizations (one for each target) and the weight of the observations varies for each target, whereas HER performs only one optimization for each one of the aggregation equations, obtaining a global set of weights which are kept fixed for the classes. Additionally, OK weights can reach extreme values
(negative or greater than 1), which on the one hand it is a useful characteristic for reduce redundancy and predict values outside the range of the data (Goovaerts, 1997, p.176), but on the other hand can lead to unacceptable results, such as negative concentrations (Goovaerts, 1997, p. 174-177) and negative kriging variances (Manchuk and Deutsch, 2007), while HER weights are limited to a range of [0,1].

Moreover, HER is flexible in the way it aggregates the probability distributions, not being necessarily a linear estimator as
OK. In terms of number of observations, being a non-parametric method, HER requires data to extract the spatial dependence structure, while OK can fit a mathematical equation with fewer data points. The mathematical function of the theoretical variogram provides an advantage in respect to computational effort. Nevertheless, relying on fitted functions can mask the lack of observations, since it still produces attractive but not necessarily reliable maps (Oliver and Webster, 2014).



## 5 Summary and conclusion

This paper presented a procedure which combines statistical learning and geostatistics which aims at overcoming parametrization and uncertainty tradeoffs present in many existing methods for spatial interpolation. For this purpose, we proposed a new spatial interpolator which is free of normality assumptions, covariance fitting, and parametrization of distributions for uncertainty estimation. Histogram via entropy reduction (HER) is designed to globally minimize the predictive uncertainty expressed by relative entropy (Kullback-Leibler divergence) between the observation and prediction. More

specifically, HER combines measures of information theory with probability aggregation methods for quantifying the available information in the dataset, extracting the structure of the data spatial correlation, relaxing normality assumptions, minimizing the uncertainty of the predictions, and combining probabilities.

Three aggregation methods (AND, OR, AND/OR) were analyzed in terms of uncertainty, and resulted in predictions ranging from conservative to more confident ones. HER's performance was also compared to popular interpolators (nearest neighbors,

inverse distance weighting, and ordinary kriging). All methods were tested under the same conditions. HER and Ordinary Kriging (OK) turned out to be the most accurate methods for different sample sizes and field types. In contrast to OK, HER featured some advantages: i) it is non-parametric, in the sense that predictions are directly based on empirical probability, thus bypassing the usual steps of variogram fitting done in OK and therefore avoiding the risk of adding information not available on the data (or losing available information); ii) it is robust against user decisions, i.e., the choice of a theoretical variogram

for OK can strongly influences the predicted $z$ values, while HER is less sensitive to the aggregation method for prediction $z$, and the bin width and distance class definitions for predicting PMF (since it does not change the spatial dependence structure, expressed in the model by $\Delta z$ PMFs and the infogram, which comes directly from the available data); iii) it allows to incorporate different uncertainty properties according to the dataset and user interest by selecting the aggregation method; iv) for uncertainty maps, HER considers not only the spatial configuration of the data, but also the field values, while kriging

variance depends on spatial data geometry only (Goovaerts, 1997, p.181); v) it is flexible to increase the number of parameters to be optimized, according to the size of the available dataset, while OK has the number of parameters fixed according to the theoretical variogram. On the other hand, being a non-parametric model, HER requires longer runtime and sufficient data to learn the spatial dependence from the data.

Considering that the quantification and analysis of uncertainties is important in all cases where maps and models of uncertain

properties are the basis for further decisions (Wellmann, 2013), HER proved to be a suitable method for uncertainty estimation, where information theoretic measures and aggregation method concepts are put together to bring more flexibility to uncertainty prediction and analysis. Additional investigation is required to analyze the method in the face of multiple-point geostatistics, spatio-temporal domains, probability and uncertainties maps, sampling designs, and handling and analyzing additional observed variables (co-variates), all of which are possible topics to be explored in future studies.



## 6 Data availability

The source code for an implementation of HER, containing spatial characterization, convex optimization and PMF prediction is published alongside this manuscript via GitHub at https://github.com/KIT-HYD/HER. The repository also includes scripts to exemplify the use of the functions and the dataset used in the case study. The synthetic field generator using Gaussian Process is available in scikit-learn (Pedregosa et al., 2011), while the code producing the fields can be found at: https://github.com/mmaelicke/random_fields.

## 7 Author contribution

ST and UE directly contributed to the design of the method and test application, to the analysis of the performed simulations, and wrote the manuscript. MM programmed the algorithm of data generation and, together with ST, calibrated the benchmark models. ST implemented the HER algorithm, performed the simulations, calibration validation design, parameter optimization, benchmarking and data support analyses. UE implemented the calculation of information theory measures, multivariate histograms operations and, together with ST and DV, the PMF aggregation functions. UE and DV contributed with interpretations and technical improvement of the model. FW provided crucial contributions to the PMF aggregation and uncertainty interpretations. DV improved the computational performance of the algorithm, implemented the convex optimization for the PMF weights, and provided insightful contributions to the method and the manuscript.

## 8 Competing interests

The authors declare that they have no conflict of interest.

## 9 Acknowledgments

The authors acknowledge support by the Deutsche Forschungsgemeinschaft (DFG) and the Open Access Publishing Fund of Karlsruhe Institute of Technology (KIT) and, for the first author, by the Graduate Funding from the German States program (Landesgraduiertenförderung).

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





**Table 1: Summary statistics and model performance of LR1-600.**

| Test set predicted by | | HER AND/OR (1) | HER pure AND (2) | HER pure OR (3) | True test set |
|---|---|---|---|---|---|
| **Summary statistics of mean predicted values of z** | mean | -0.98 | -0.98 | -0.98 | -1.00 |
| | standard deviation | 0.89 | 0.89 | 0.90 | 1.03 |
| | entropy ($H$) | 4.07 | 4.04 | 4.10 | 4.39 |
| | maximum | 1.32 | 1.26 | 1.33 | 2.14 |
| | median | -0.83 | -0.82 | -0.85 | -0.96 |
| | minimum | -2.82 | -2.77 | -2.92 | -3.75 |
| | kurtosis | 2.23 | 2.19 | 2.27 | 2.44 |
| | skewness | 0.02 | 0.02 | 0.03 | 0.02 |
| **Summary statistics of predicted z PMF** | median entropy | 3.45 | 3.75 | 4.17 | – |
| | $z$ maximum[1] | 2.40 | 3.20 | 2.60 | – |
| | $z$ minimum[1] | -4.20 | -7.00 | -4.80 | – |
| | target (49,73): [95% CI] | [-0.40, 1.60] | [-0.60, 1.60] | [-1.20, 2.20] | – |
| | mean | 0.69 | 0.66 | 0.70 | 1.35 |
| | target (47,16): [95% CI] | [-2.00, -0.20] | [-2.20, 0.00] | [-2.60, 0.20] | – |
| | mean | -0.99 | -1.00 | -0.98 | -1.02 |
| | target (25,63): [95% CI] | [-2.40, -0.40] | [-2.40, -0.40] | [-4.00, 0.60] | – |
| | mean | -1.19 | -1.33 | 1.20 | -1.34 |
| | target (10,42): [95% CI] | [-3.00, -1.20] | [-3.20, -1.20] | [-3.80, -0.80] | – |
| | mean | -2.06 | -2.06 | -2.05 | -1.64 |
| **Performance** | $E_{MA}$ | 0.43 | 0.43 | 0.44 | – |
| | $E_{NS}$ | 0.72 | 0.72 | 0.71 | – |
| | $D_{KL}$ | 3.54 | 3.58 | 3.76 | – |

[1] Considering a 95% confidence interval.
CI: confidence interval.





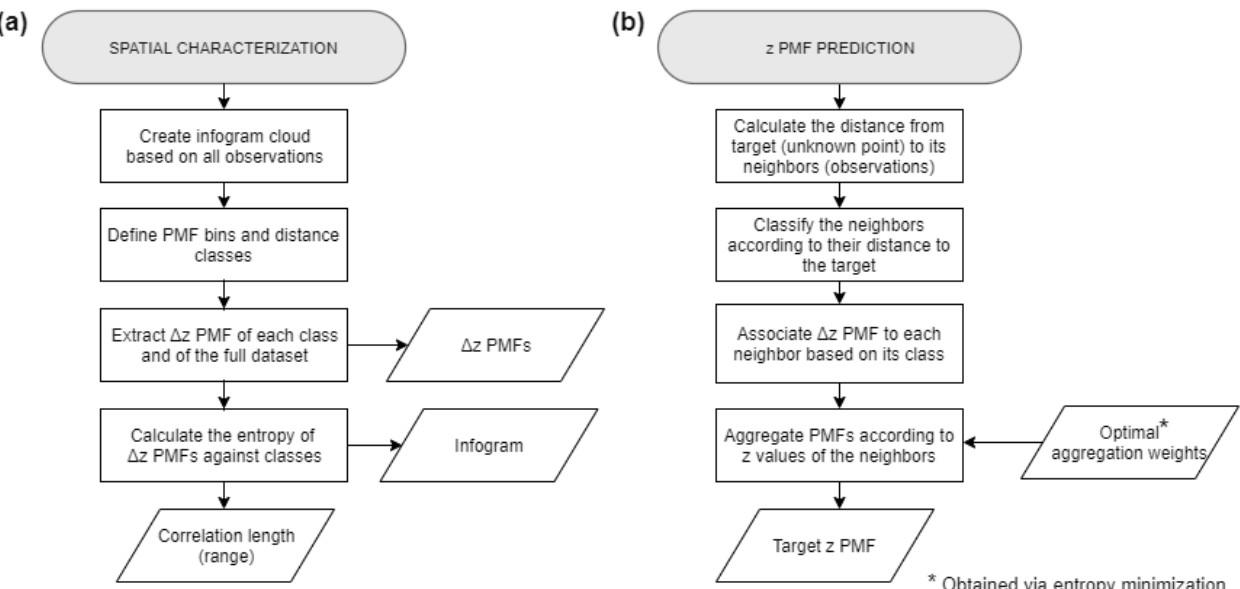

**Figure 1: HER method. Flowcharts illustrating the a) spatial characterization, and b) probability mass functions (PMF) prediction.**






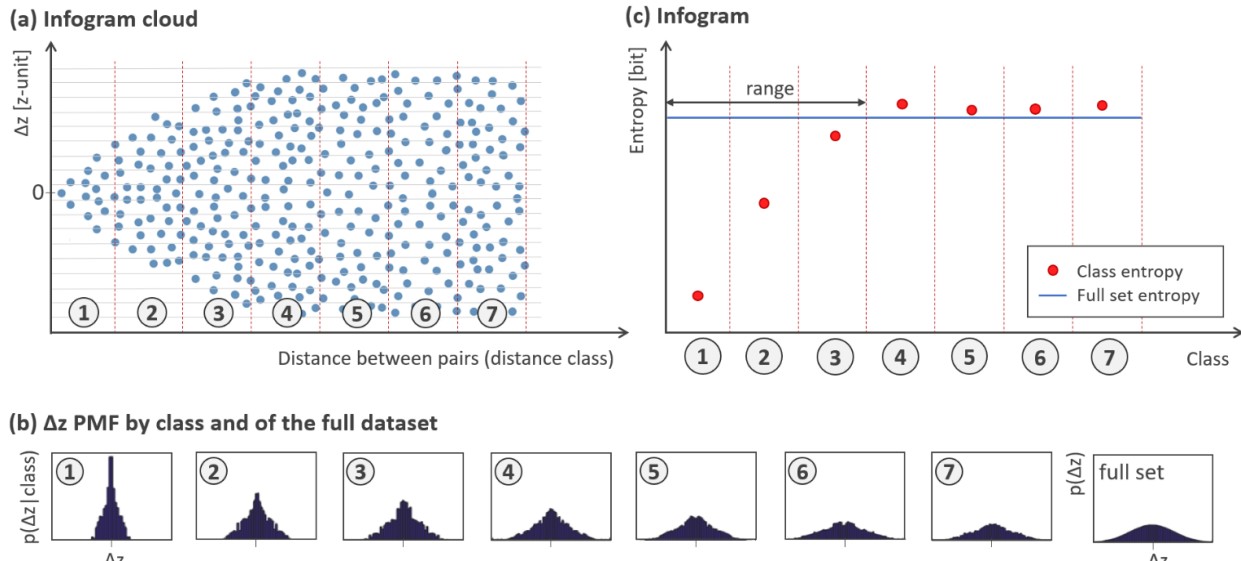

**Figure 2: Spatial characterization. Illustration of the a) infogram cloud, b) *Δz* PMFs, and c) infogram.**





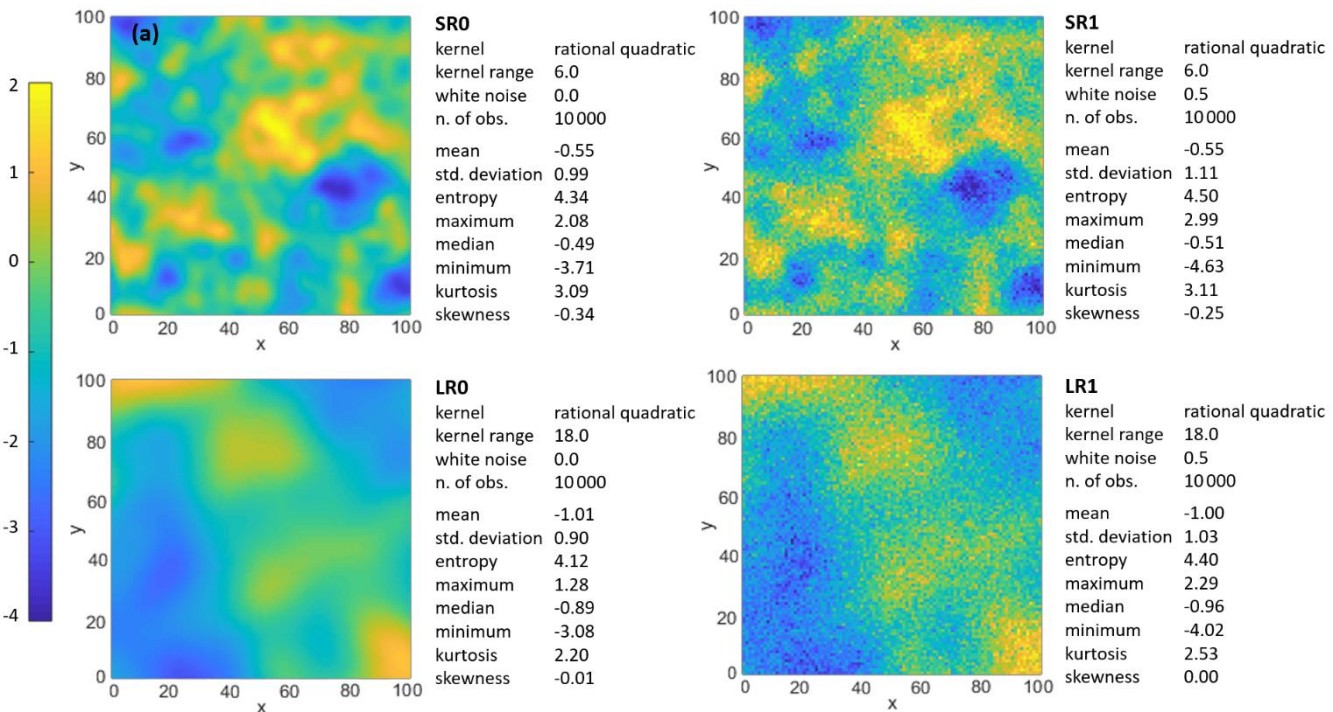

**Figure 3: Synthetic fields and summary statistics: a) SR0, b) SR1, c) LR0, and d) LR1.**





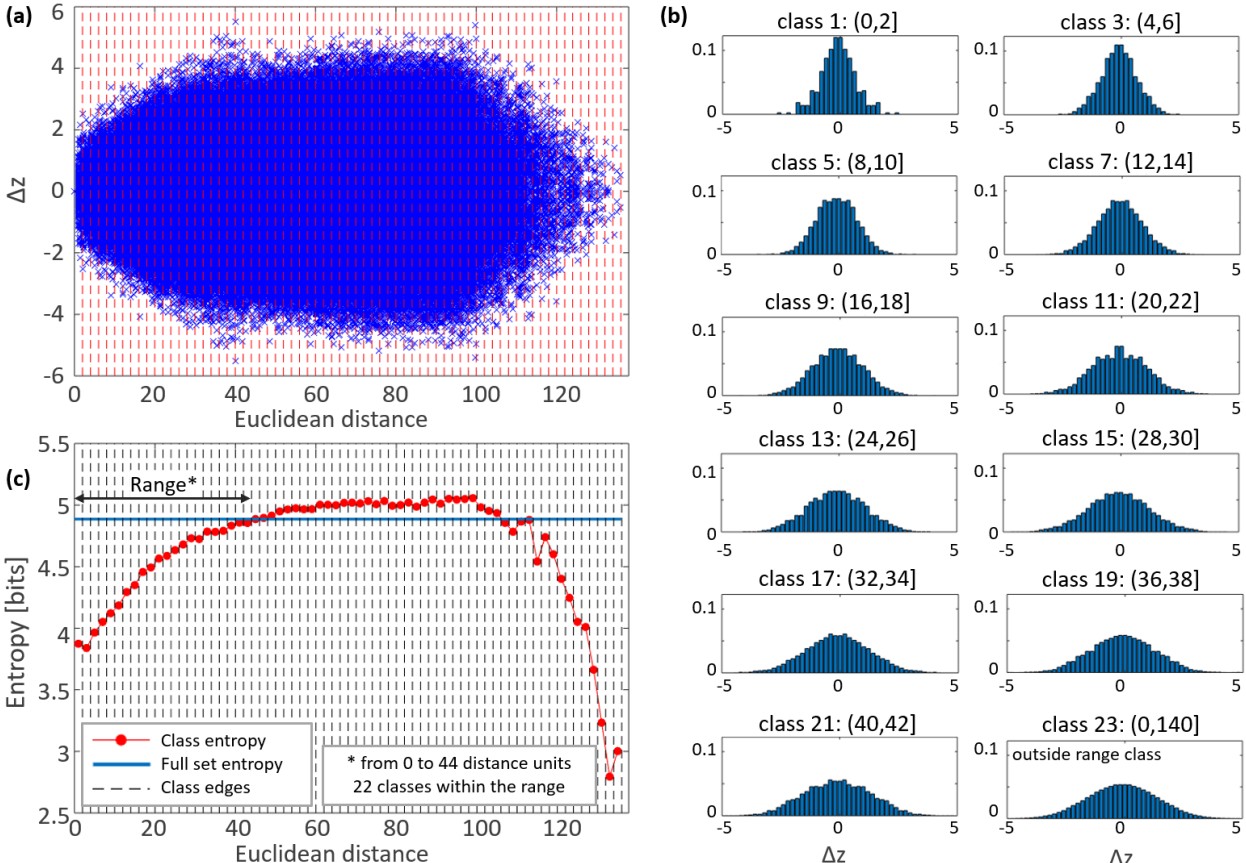

**Figure 4: Spatial characterization of LR1-600: a) infogram cloud, b) Δz PMFs by class, and c) infogram.**





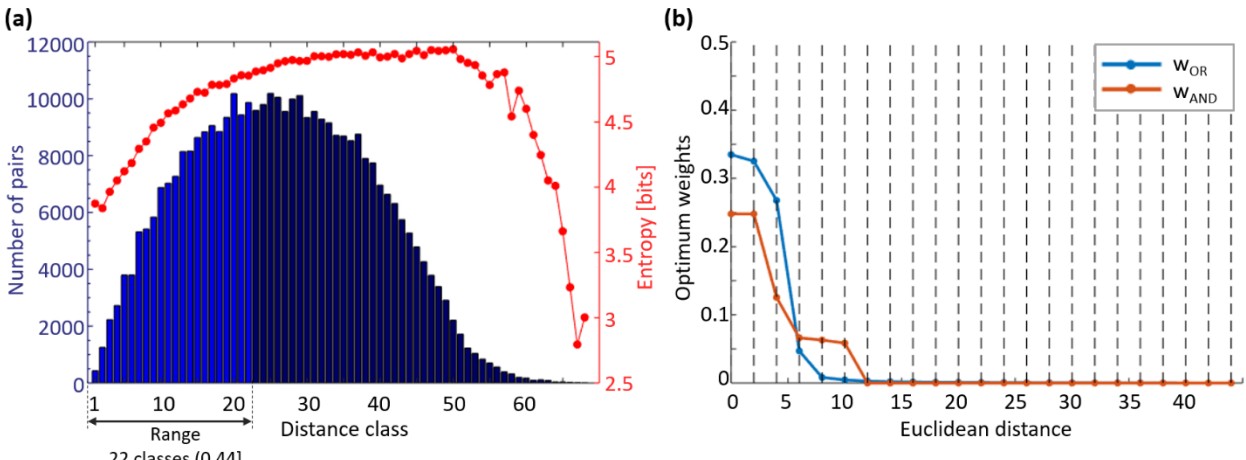


**Figure 5: LR1-600: a) class cardinality, and b) optimum weights, Eqs. (4) and (5).**




**Figure 6: LR1-600 results: a) predicted mean of *z*, b) entropy map (bits), and c) *z* PMF prediction for selected points. The first, second and third columns of the panel refer to the results of model 1 (AND/OR), model 2 (AND), and model 3 (OR), respectively.**



**Figure 7: Performance comparison of NN, IDS, OK and HER: a,b) mean absolute error, c,d) Nash–Sutcliffe efficiency, and e,f) Kullback Leibler divergence scoring rule, for the SR datasets in the left column and the LR datasets in the right. Continuous line**
**refers to datasets without noise and dashed lines to datasets with noise.**