# Peer review of "HER: an information theoretic alternative for geostatistics"

_Hydrology and Earth System Sciences, 2020_

## Referee Comment (RC1) · Anonymous Referee #1 · 17 Mar 2020

Dear Authors,

Thank you very much for this work.

I think the work is very interesting but the paper requires some reviews before it can be published.

My main concerns are with respect to the structure of the paper. In its current status, I think the sections are unbalanced in terms of the length and some of them are mixed. E.g. in section 5. Summary and conclusions, the section is missed with the discussion – also i would suggest not including references in the conclusions.

I would suggest to shorten the paper by 1) stating clearly the messages in the paragraphs, 2) rephrasing unclear wording (e.g. lines 31-33, 37-43, etc), 3) removing un-

necessarily wording (e.g. 87-88 "This section is based on..."-> see detains in Cover and thomas, 2006) and/or, 4) including necessary explanations (e.g. line 101, i suggest remove the word "best" or include from what point of view is "best").

I also recommend using math notation consistently, e.g. bold for vectors, capital for random variables, etc.

Please, also: - add references for nearest neighbors, and inverse distance weighting (see line 25), - use complete names before using abbreviations (e.g. Sect.2 – the full name should be given before providing any abbreviation), - have a look at the typos (there are many), and - maybe, good to briefly explain what is the "infogram cloud" - i am not sure that all readers are familiar with it.

In case that the editor asks a revised version of the the manuscript, i am very happy to serve as reviewer of the revised version.

Once again, thank you very much for this work.

Kind Regards, Reviewer

---

## Referee Comment (RC2) · Anonymous Referee #2 · 28 Apr 2020

This paper presents a method called Histogram via Entropy Reduction (HER) for the interpolation of spatial geophysical data. This method is based on information theory measures of entropy and relative entropy, and has advantages over benchmarks of kriging and nearest neighbor methods in terms of its generality and lack of assumptions. The authors present the methodology which determines spatial dependency structure based on observed data points, and estimates optimal weighting parameters used to predict a given variable at a location. An application to several synthetic datasets shows high effectiveness of the method relative to three existing interpolation techniques.

Overall this paper was interesting and clearly written, and the figures are very informative and help to illustrate complex concepts. Although I do not have a background

in geostatistics or interpolation of sparse datasets, this paper seems to introduce a promising avenue of how IT measures can be advantageous in this field. Some comments and suggestions listed below, which consist of minor revisions/technical corrections. They mainly highlight places that could use additional explanation or clarification.

Main Comments: Line 111: on the description of creating the "infogram cloud": at first it was not clear to me whether an "infogram" was an existing technique that I was unfamiliar with, or designed by the authors. I think it is the latter (based on line 134) – either way, this aspect could be made more clear earlier in the subsection, that you have developed this graphical technique called an infogram that shows spatial correlation structure.

Line 145: Could you add a bit more information on what effect/advantage this has? It seems like attributing a small probability to every category would make a larger difference to some types of distributions than to others.

Section 2.3.1: For readers less familiar with aggregating probabilities towards a spatial context, I this description could benefit from some sort of illustration or simple example that shows the difference between the different pooling operators in Eqs 4-7. For example, show two measurement points D1 and D2 with a target location A somewhere between them, and show how the measures differ. This actually become more clear to me with the later discussion in Line 445 onward, so maybe some of these aspects could be brought forward earlier.

Section 3.1: Are there implications to model performance of the Gaussian process used to make the test cases? I wonder how this would compare to a realistic landscape (or whether this is considered very close to a "real world" case). Something is mentioned later about this in the discussion regarding the OK method, but more information would be beneficial here.

Line 360: I think this is explaining why the infogram illustration in Figure 2a is very different from that in Figure 4a for the farther distances (which is because with the

data, there are fewer pairs that exist at those farthest distances). If this is correctly interpreted, it could be brought up in the description of Figure 2a and the infogram cloud-shape in general.

Technical/writing style comments: Abstract: There are several sentences here with parenthesis for additional context, I would recommend re-writing these without as many parentheses to potentially simplify and help the flow.

Line 38: applying

Line 90: H(X) is upper bounded by infinity in a continuous case, but as you mention in the next sentence and the equation that this case is discrete – the upper bound should be log2(N) where N is the number of bins or categories. Figure 6: It is hard to see the targets in a few of the maps, I think because the markers are in the background behind the observation markers. It would help to make outlines of the marker shapes bolder or colored to show which target is which.

Line 379: "differentiate between", instead of differ? Line 429: I found this sentence to be unecessary

---

## Author Response (AR1)

Dear Dr. Kelleher and referees,

The current document consolidates the referee comments, the author responses posted so far, and the actions taken in the major revision of the manuscript. It is organized according to the following color code:

- Black: Referee comments
- Blue: Authors' responses from the previous stage
- Red: Authors' major revisions and responses for the new version of the manuscript

After this section of comments and responses, you can find the manuscript pdf with the tracked changes. Please note that the page numbers and lines in our current responses (red) refer to the revised, tracking-free version of the manuscript.

The new version of the manuscript contains the adjustments requested by the referees during this open discussion, summarized below:

1. Adding a paragraph:
   a. with a practical example of the three aggregation methods (Sect. 2.3.1)
   b. discussing real-world data and GP (Sect. 4.3)
   c. discussing undesired uncertainty when predicting at measurement locations (Sect. 4.3)
   d. discussing redundant measurements (Sect. 4.3)
2. Making the method clearer (e.g., infogram cloud, infogram, etc.)
3. Reviewing and restructuring discussion(Sect. 4.3) and conclusion (Sect. 5)
4. Reviewing math notation
5. Adding Dr. Ralf Loritz as co-author
6. Reviewing all figures (300 dpi quality, readability, and consistency)
7. Improving text readability and fluidity

Best regards,

Stephanie Thiesen, Diego M. Vieira, Mirko Mälicke, Ralf Loritz, J. Florian Wellmann, and Uwe Ehret

**Referee #1**

Dear Authors,

Thank you very much for this work. I think the work is very interesting but the paper requires some reviews before it can be published.

My main concerns are with respect to the structure of the paper. In its current status, I think the sections are unbalanced in terms of the length and some of them are mixed. E.g. in section 5. Summary and conclusions, the section is missed with the discussion – also i would suggest not including references in the conclusions.

I would suggest to shorten the paper by 1) stating clearly the messages in the paragraphs, 2) rephrasing unclear wording (e.g. lines 31-33, 37-43, etc), 3) removing un necessarily wording (e.g. 87-88 "This section

is based on. . ."-> see detains in Cover and thomas, 2006) and/or, 4) including necessary explanations (e.g. line 101, i suggest remove the word "best" or include from what point of view is "best").

I also recommend using math notation consistently, e.g. bold for vectors, capital for random variables, etc.

Please, also:

- add references for nearest neighbors, and inverse distance weighting (see line 25),
- use complete names before using abbreviations (e.g. Sect.2 – the full name should be given before providing any abbreviation),
- have a look at the typos (there are many), and - maybe, good to briefly explain what is the "infogram cloud" - i am not sure that all readers are familiar with it.

In case that the editor asks a revised version of the the manuscript, i am very happy to serve as reviewer of the revised version. Once again, thank you very much for this work.

Kind Regards,

Reviewer

**Response:** We thank referee #1 for reviewing our manuscript and providing his/her feedback. Since the recommendations encompass a more general aspect, the authors will consider all suggestions (review paper structure, math notation, and abbreviations, insert/exclude references, include clarifications) during a textual revision of the manuscript.
We revised and restructured the discussion (Sect. 4.3) and the conclusion (Sect. 5) sections. Furthermore, we proceeded with all specific suggestions (where the line number was stated by the referee) and endeavored to follow all his/her general suggestions, i.e, a thorough revision of the manuscript considering: i) math notation, ii) misplaced references, iii) unbalanced sections, iv) unclear abbreviations, v) missing references, and vi)  lack of clarity.

**Referee #2**

This paper presents a method called Histogram via Entropy Reduction (HER) for the interpolation of spatial geophysical data. This method is based on information theory measures of entropy and relative entropy, and has advantages over benchmarks of kriging and nearest neighbor methods in terms of its generality and lack of assumptions. The authors present the methodology which determines spatial dependency structure based on observed data points, and estimates optimal weighting parameters used to predict a given variable at a location. An application to several synthetic datasets shows high effectiveness of the method relative to three existing interpolation techniques.

Overall this paper was interesting and clearly written, and the figures are very informative and help to illustrate complex concepts. Although I do not have a background in geostatistics or interpolation of sparse datasets, this paper seems to introduce a promising avenue of how IT measures can be advantageous in this field. Some comments and suggestions listed below, which consist of minor

revisions/technical corrections. They mainly highlight places that could use additional explanation or clarification.

Main Comments:

**Comment 1:** Line 111: on the description of creating the "infogram cloud": at first it was not clear to me whether an "infogram" was an existing technique that I was unfamiliar with, or designed by the authors. I think it is the latter (based on line 134) – either way, this aspect could be made more clear earlier in the subsection, that you have developed this graphical technique called an infogram that shows spatial correlation structure.

**Response 1:** Indeed, the term "infogram cloud" in L.111 is out of place. The same observation was made by referee #1. To avoid early questions, we propose rephrasing the sentence and including the name of the 3 assets used for the spatial characterization (Infogram cloud, Δz PMFs, and Infogram) when they first appear (L.112-121) as follows:

"As shown in Fig. 1a, the spatial characterization phase aims to obtain: Δz probability mass functions (PMFs), where z is the variable under study; the behavior of entropy as a function of lag distance (which the authors denominate 'infogram'); and, finally, the correlation length (range). These outputs are outlined in Fig. 2 and attained in the following steps: i. Infogram cloud (Fig. 2a): [...] ii. Δz PMFs (Fig. 2b): [...] iii. Infogram (Fig. 2c)".

As previously proposed, we rephrased the sentence and improved the explanation regarding infogram cloud (Sect. 2.2, p. 4, l. 102-115).

**Comment 2:** Line 145: Could you add a bit more information on what effect/advantage this has? It seems like attributing a small probability to every category would make a larger difference to some types of distributions than to others.

**Response 2:** For the application of HER (mainly when using the log-linear aggregation method), it is desirable to assure that all bins of the distribution have a nonzero probability. This guarantees that there is always an intersection when aggregating PMFs. In this way, when the intersection between two PMFs happens only on the previously empty bins, the resulting PMF is a uniform distribution, i.e., the method effectively applies a maximum-entropy approach.

In addition, Darscheid et al. (2018) checked the impact of five alternatives for nonzero probability to a range of typical distributions (uniform, Dirac, normal, multimodal, and irregular) and concluded that, for the cases where no distribution is known a priori, three methods (including the one used in the paper) performed well across analyzed distributions. In order to add more information regarding the nonzero probability, we suggest rewriting the paragraph as follows:

"[...] The bin size was defined based on Thiesen et al. (2018), by comparing the cross entropy $(H_{pq}=H(p)+D_{KL}(p||q))$ between the full learning set and subsamples for various bin widths. The selected one shows a stabilization of the cross entropy for small sample sizes, meaning that the bin size is reasonable for small and large sample sizes and analyzed distribution shapes. For favoring comparability, the bins were kept the same for all applications and performance calculations.

*Additionally, to avoid distributions with empty bins which might make the PMF combination (presented in Sect. 2.3.1) unfeasible, as recommended by Darscheid et al. (2018), we assigned a small probability equivalent to the probability of a single point-pair count to all bins in the histogram after converting it to a PMF by normalization. This guarantees that there is always an intersection when aggregating PMFs, and that we obtain a uniform distribution (maximum-entropy) in case we multiply distributions where the overlap happens uniquely on the previously empty bins. Furthermore, as shown in the Darscheid et al. (2018) study, for the cases where no distribution is known a priori, adding one counter to each empty bin performed well across different distributions."*

We added a brief discussion of the nonzero probabilities and bin selection in the revised version of the manuscript (Sect. 2.2, p. 5, l. 145-152).

**Comment 3:** Section 2.3.1: For readers less familiar with aggregating probabilities towards a spatial context, I this description could benefit from some sort of illustration or simple example that shows the difference between the different pooling operators in Eqs 4-7. For example, show two measurement points D1 and D2 with a target location A somewhere between them, and show how the measures differ. This actually become more clear to me with the later discussion in Line 445 onward, so maybe some of these aspects could be brought forward earlier.

**Response 3:** The authors agree that an illustration of the aggregation methods could be beneficial for showing the practical meaning of each one of the aggregation options later explored in the application case. Since the example in the spatial context is given (without illustration) in L.185-201 and L.445-448, we believe that we could explore the practical implication of the methods in the end of the section 2.3.1. We estimate that it will increase the size of the paper in half page (figure plus brief explanation). A preview of the additional figure and explanation is shown below.

*"The practical differences between the pooling operators used in this paper are illustrated in Fig. 3, where Fig. 3a introduces the two PMFs to be combined, and Figs. 3b,c,d show the resulting PMFs for Eqs. (5), (4), and (7), respectively. In Fig. 3b, we use unitary PMF weights, so that the multiplication of the bins (AND aggregation) leads to a simple intersection of PMFs weighted by the bin height. In Fig. 3c, we use equal weights to both PMFs, and the resulting distribution is the arithmetic average of the bin probabilities. Fig. 3d shows a log-linear aggregation of the two previous distributions (Figs. 3b,c). In all three cases, if the weight of one distribution is set to one and the other is set to zero (not shown), the resulting PMF would be equal to the distribution which receives all the weight. Specifically for Eq. (7), this means that the final distribution may result in a pure AND, Eq. (5), or pure OR aggregation, Eq. (4), as special cases."*

[Figure]

Figure 3: Examples of the different pooling operators. a) Normal PMFs $N(\mu,\sigma^2)$ to be combined; b) log-linear aggregation, Eq. (5); c) linear aggregation, Eq. (4); and d) log-linear aggregation of (b) and (c), Eq. (7).

We included a practical example (figure + discussion) of the three aggregation methods in the revised version of the manuscript (Sect. 2.3.1, p. 8, l. 228-234 + Figure 3 p.28).

**Comment 4:** Section 3.1: Are there implications to model performance of the Gaussian process used to make the test cases? I wonder how this would compare to a realistic landscape (or whether this is considered very close to a "real world" case). Something is mentioned later about this in the discussion regarding the OK method, but more information would be beneficial here.

**Response 4:** Thanks for raising this question. The purpose of the paper is to test HER and demonstrate its performance in face of an established geostatistical method, namely OK. Thus, for testing HER, a field generated by Gaussian Process (GP) enables us to have a controlled dataset where we could examine their performance in fields with different characteristics (short and long range, with and without noise, small or large sample size). Since GP datasets fulfill the assumptions of Ordinary Kriging, it allows a fair comparison between the methods. We can say that GP and OK are the inverse of each other. While GP generates a dataset which follows a multivariate gaussian distribution with a known covariance function, OK estimates the stochastic process behind a dataset by fitting a variogram (or covariance function) and assuming that the residuals (i.e., estimation error) follow a gaussian distribution (Kitanidis, 1997, p.95).

It is true that a real-world dataset may not necessarily have the gaussianity properties given by the GP. Therefore it is the role of the geostatistician to guarantee that the data fulfill the method assumptions. . When it is not the case, e.g., it is common to transform the data so that it fits the assumptions, and back-transform it in the end. It is worth mentioning that, while developing the manuscript, the authors tested HER in a real-world case of digital elevation model data. Although HER and OK both performed well, its inclusion would require a proper geostatistical description of the dataset, which would be out of the scope of this paper and therefore we discarded its presentation to keep the paper as short as possible and because GP covered a broader scope of fields. Altogether, this  means that the test proposed using GP is also related to a real-world problem.

The authors understand that, due to being non-parametric, HER can deal with different data properties without the need of transforming the available data. HER does not require fitting of a theoretical function for extracting the spatial correlation, because its spatial dependence structure is derived directly from the

available data. And since HER uses binned transformation of the data, it is also possible to handle binary (e.g. contaminated and safe areas) or even, with small adaptations, handle categorical data (soil types), covering another spectrum of real data.

Although parts of the above discussion were mentioned in L.470 and L.484, the authors believe that it would be beneficial to review the discussion section (Sect. 4.3) to include the above arguments in the manuscript.

We included the previous discussion regarding real-word data and implications of using GP in the revised version of the manuscript (Sect. 4.3.2, p. 17, l. 486-492).

**Comment 5:** Line 360: I think this is explaining why the infogram illustration in Figure 2a is very different from that in Figure 4a for the farther distances (which is because with the data, there are fewer pairs that exist at those farthest distances). If this is correctly interpreted, it could be brought up in the description of Figure 2a and the infogram cloud-shape in general.

**Response 5:** Thanks for pointing it out. As Fig. 2 as a whole merely illustrates what one could expect to get from the spatial characterization part, we decided to show basically the behavior of the first distance classes, where the spatial correlation is stronger. "Ignoring" the last classes is a common practice when analyzing spatial correlation, when the geostatistician defines a distance cutoff (maximum lag) for their analysis. Thus, considering that this omission, although discussed in Fig.4a, is in principle expected, the authors suggest including in L.129 the following clarification "*Note that in the illustrative case of Fig. 2, we limited the number of classes shown to four classes beyond the range. A complete infogram cloud and infogram is presented and discussed in the method application, Fig. 4.*"

We included the previous clarification in the revised version of the manuscript (Sect. 2.2, p. 5, l. 123-125).

Technical/writing style comments:

**Comment 6:** Abstract: There are several sentences here with parenthesis for additional context, I would recommend re-writing these without as many parentheses to potentially simplify and help the flow.

**Response 6:** Thanks. We will adapt the writing style in a revised version of the manuscript considering this point. The authors restructured the whole abstract.

**Comment 7:** Line 38: applying

**Response 7:** Ok, thanks. Adjusted.

**Comment 8:** Line 90: H(X) is upper bounded by infinity in a continuous case, but as you mention in the next sentence and the equation that this case is discrete – the upper bound should be log2(N) where N is the number of bins or categories.

**Response 8:** We agree that it can cause misunderstanding, we will refine this paragraph. The authors adjusted the paragraph considering entropy in discrete distributions (Sect. 2.1, p. 3, l. 85-86).

**Comment 9:** Figure 6: It is hard to see the targets in a few of the maps, I think because the markers are in the background behind the observation markers. It would help to make outlines of the marker shapes bolder or colored to show which target is which.

**Response 9:** Thanks. We will endeavor to improve the visibility of the points. The authors adjusted the previous Figure 6 (Figure 7, p.32).

**Comment 10:** Line 379: "differentiate between", instead of differ?
**Response 10:** Ok, thanks. Adjusted.

**Comment 11:** Line 429: I found this sentence to be unecessary
**Response 11:** Thanks. We will consider removing it in the paper writing style revision. Sentence removed.

**To the editor**

We also suggest including in the discussion section (Sect. 4.3) two issues noticed by the authors during the revision process and while testing the method for assessing data uncertainty. The first one is that, since the dataset was evenly spaced, a possible issue of redundant information in case of clustered samples was not considered. Another matter that we wish to briefly discuss and propose theoretical solutions to is that, depending on how the first distance class is chosen, HER can lead to undesired uncertainty when predicting the value at the observations themselves.

We included a brief discussion and proposed theoretical solutions for undesired uncertainty and redundancy issues in the revised version of the manuscript (Sect. 4.3.3, p. 18, l. 518-523 & Sect. 4.3.4, p. 18, l. 531-537).

In addition, we would also like to review the manuscript including some terminologies which are more precise to describe the method and its implications, and they could assist to foster the proper search for scholarly literature, mainly: E-type estimate (for the expected value obtained using HER) and conditional distribution (for the results of the aggregation method and PMFs obtained with HER). Although these terms are implicit in the method and explained, the authors would like to include them explicitly.

The mentioned terminologies were explicitly mentioned throughout the new version of the manuscript.

[revised manuscript text omitted]

**(Model 1)** $P_{G_{AND}}(z_0)^{\alpha} \cdot P_{G_{OR}}(z_0)^{\beta}$  **(Model 2)** $P_{G_{AND}}(z_0)$  **(Model 3)** $P_{G_{OR}}(z_0)$

● target A (10,42)   ● target B (25,63)   ◆ target C (47,16)   ◆ target D (49,73)   + calibration set

**(c.1)** target A (10,42) H = 3.31 bits   target B (25,63) H = 3.42 bits
target C (47,16) H = 3.25 bits   target D (49,73) H = 3.52 bits

**(c.2)** target A (10,42) H = 3.58 bits   target B (25,63) H = 3.52 bits
target C (47,16) H = 3.60 bits   target D (49,73) H = 3.73 bits

**(c.3)** target A (10,42) H = 4.04 bits   target B (25,63) H = 4.68 bits
target C (47,16) H = 4.01 bits   target D (49,73) H = 4.29 bits

[Figure]

**Figure 7:** LR1-600 results: a)  E-type estimate of *z*, b) entropy map (bit), and c) *z*- PMF prediction for selected points. The first, second and third columns of the panel refer to the results of model 1 (AND/OR), model 2 (AND), and model 3 (OR), respectively.

840

[Figure]

845

[Figure]

**Figure 8: Performance comparison of NN, IDS, OK and HER: a,b) mean absolute error, c,d) Nash–Sutcliffe efficiency, and e,f) Kullback-Leibler divergence scoring rule, for the SR datasets in the left column and the LR datasets in the right. Continuous line refers to datasets without noise and dashed lines to datasets with noise.**

850

**HER: an information theoretic alternative for geostatistics**

Stephanie Thiesen[1], Diego M. Vieira[2,3], Mirko Mälicke[1], Ralf Loritz[1], J. Florian Wellmann[4], Uwe Ehret[1]

[1]Institute of Water Resources and River Basin Management, Karlsruhe Institute of Technology, Karlsruhe, Germany
5   [2]Department for Microsystems Engineering, University of Freiburg, Freiburg, Germany
[3]Bernstein Center Freiburg, University of Freiburg, Freiburg, Germany
[4]Computational Geosciences and Reservoir Engineering, RWTH Aachen University, Aachen, Germany

**Supplement S1: Summary statistics of the resampled datasets**

10   Table S1.1 and Table S1.2 summarize the statistics of the learning, validation, test, and full datasets.

**Table S1.1: Summary statistics of the resampled datasets – Short-range dataset (SR0 and SR1).**

| Sample size | 200 | 400 | 600 | 800 | 1000 | 1500 | 2000 | 2000 (val. set) | 2000 (test set) | 10 000 (full set) |
|---|---|---|---|---|---|---|---|---|---|---|
| **SR0** | | | | | | | | | | |
| mean | -0.57 | -0.59 | -0.58 | -0.59 | -0.59 | -0.58 | -0.57 | -0.53 | -0.56 | -0.55 |
| sd. | 1.05 | 1.06 | 1.02 | 1.01 | 0.99 | 0.99 | 0.99 | 0.99 | 1.00 | 0.99 |
| $H$ | 4.27 | 4.38 | 4.34 | 4.33 | 4.31 | 4.32 | 4.32 | 4.31 | 4.34 | 4.34 |
| max. | 1.76 | 1.92 | 1.92 | 1.92 | 1.92 | 1.92 | 2.05 | 2.08 | 2.02 | 2.08 |
| median | -0.42 | -0.50 | -0.51 | -0.56 | -0.54 | -0.52 | -0.52 | -0.46 | -0.50 | -0.49 |
| min. | -3.68 | -3.68 | -3.68 | -3.68 | -3.68 | -3.68 | -3.68 | -3.67 | -3.71 | -3.71 |
| kur. | 3.21 | 3.04 | 3.12 | 3.15 | 3.17 | 3.14 | 3.12 | 3.18 | 3.07 | 3.09 |
| sk. | -0.62 | -0.43 | -0.41 | -0.35 | -0.35 | -0.32 | -0.30 | -0.36 | -0.33 | -0.34 |
| **SR1** | | | | | | | | | | |
| mean | -0.52 | -0.54 | -0.55 | -0.57 | -0.57 | -0.57 | -0.56 | -0.54 | -0.54 | -0.55 |
| sd. | 1.17 | 1.17 | 1.14 | 1.12 | 1.11 | 1.10 | 1.10 | 1.11 | 1.12 | 1.11 |
| $H$ | 4.46 | 4.54 | 4.51 | 4.50 | 4.49 | 4.49 | 4.49 | 4.49 | 4.52 | 4.50 |
| max. | 2.50 | 2.70 | 2.70 | 2.70 | 2.70 | 2.70 | 2.99 | 2.96 | 2.86 | 2.99 |
| median | -0.36 | -0.51 | -0.51 | -0.55 | -0.56 | -0.54 | -0.53 | -0.51 | -0.48 | -0.51 |
| min. | -3.66 | -3.66 | -3.66 | -3.84 | -3.84 | -4.01 | -4.01 | -4.63 | -4.25 | -4.63 |
| kur. | 2.82 | 2.83 | 2.93 | 2.94 | 2.99 | 3.03 | 3.04 | 3.24 | 3.09 | 3.11 |
| sk. | -0.40 | -0.15 | -0.19 | -0.19 | -0.18 | -0.20 | -0.20 | -0.28 | -0.26 | -0.25 |

sd. = standard deviation; $H$ = entropy; max. = maximum; min. = minimum; kur. = kurtosis; sk. = skewness.

**Table S1.2: Summary statistics of the resampled datasets – Long-range dataset (LR0 and LR1).**

| Sample size | 200 | 400 | 600 | 800 | 1000 | 1500 | 2000 | 2000 (val. set) | 2000 (test set) | 10 000 (full set) |
|---|---|---|---|---|---|---|---|---|---|---|
| | | | | | **LR0** | | | | | |
| mean | -0.98 | -0.96 | -1.03 | -1.01 | -1.01 | -1.01 | -1.02 | -1.00 | -1.02 | -1.01 |
| sd. | 0.90 | 0.88 | 0.89 | 0.89 | 0.90 | 0.91 | 0.91 | 0.90 | 0.91 | 0.90 |
| $H$ | 3.99 | 4.02 | 4.07 | 4.09 | 4.09 | 4.11 | 4.11 | 4.11 | 4.12 | 4.12 |
| max. | 1.04 | 1.15 | 1.23 | 1.23 | 1.23 | 1.23 | 1.23 | 1.28 | 1.27 | 1.28 |
| median | -0.77 | -0.81 | -0.92 | -0.92 | -0.91 | -0.91 | -0.92 | -0.88 | -0.89 | -0.89 |
| min. | -2.78 | -2.78 | -3.07 | -3.07 | -3.07 | -3.08 | -3.08 | -3.00 | -3.07 | -3.08 |
| kur. | 2.11 | 2.18 | 2.26 | 2.24 | 2.21 | 2.16 | 2.20 | 2.22 | 2.16 | 2.20 |
| sk. | -0.09 | -0.07 | 0.02 | 0.02 | 0.03 | 0.03 | 0.03 | -0.03 | 0.00 | -0.01 |
| | | | | | **LR1** | | | | | |
| mean | -0.92 | -0.91 | -0.99 | -1.00 | -1.00 | -1.01 | -1.01 | -1.01 | -1.00 | -1.00 |
| sd. | 0.98 | 1.00 | 1.01 | 1.02 | 1.03 | 1.04 | 1.03 | 1.05 | 1.03 | 1.03 |
| $H$ | 4.21 | 4.31 | 4.34 | 4.37 | 4.38 | 4.40 | 4.39 | 4.41 | 4.39 | 4.40 |
| max. | 1.40 | 1.87 | 1.87 | 1.87 | 1.96 | 1.96 | 2.00 | 2.29 | 2.14 | 2.29 |
| median | -0.88 | -0.91 | -0.97 | -0.98 | -0.99 | -0.99 | -0.98 | -0.98 | -0.96 | -0.96 |
| min. | -3.19 | -3.65 | -3.65 | -3.74 | -3.74 | -3.74 | -3.95 | -4.02 | -3.75 | -4.02 |
| kur. | 2.51 | 2.67 | 2.56 | 2.56 | 2.59 | 2.50 | 2.53 | 2.59 | 2.44 | 2.53 |
| sk. | -0.09 | 0.02 | 0.06 | 0.04 | 0.06 | 0.05 | 0.04 | -0.02 | 0.02 | 0.00 |

sd. = standard deviation; $H$ = entropy; max. = maximum; min. = minimum; kur. = kurtosis; sk. = skewness.

**Supplement S2: Parameter tuning**

This supplement consolidates the final parameters used in the models presented in Sect. 4.2. Particularly for HER, Fig. S2.1 presents the final weights optimized for Eqs. (4) and (5). It was limited to 18 grid units (nine distance classes), due to the small contribution of the faraway classes. Similarly, Fig. S2.2 shows $\alpha$ and $\beta$ weights of Eq. (67). Finally, Table S2.1 and Table S2.2 summarize the calibrated parameters obtained for each model (varying method, sample size and dataset type).

[Figure]

20  **Figure S2.1: HER optimized weights by distance class: a,b) $w_{OR}$, Eq. (4), and c,d) $w_{AND}$. Eq. (5). SR datasets on the left panel and LR datasets on the right panel. Continuous line refers to datasets without noise and dashed lines to datasets with noise.**

[Figure]

25  **Figure S2.2. HER $\alpha$ and $\beta$ weights by sample size, Eq. (67): a) SR datasets on the left panel, and b) LR datasets on the right panel. Continuous line refers to datasets without noise and dashed lines to datasets with noise.**

**Table S2.1: Method calibration by sample size – Parameters of the models for the short-range dataset (SR0 and SR1).**

| Model sample size | 200 | 400 | 600 | 800 | 1000 | 1500 | 2000 |
|---|---|---|---|---|---|---|---|
| **Method** **Parameter**[+] | | | | SR0 | | | |
| NN n.n. | 1 | 1 | 1 | 1 | 1 | 1 | 1 |
| IDS exp. | 2 | 2 | 2 | 2 | 2 | 2 | 2 |
| OK n.n. | 12 | 12 | 12 | 12 | 12 | 12 | 12 |
| lag width | 2 | 2 | 2 | 2 | 2 | 2 | 2 |
| variogram | Spherical | Spherical | Spherical | Spherical | Spherical | Spherical | Spherical |
| eff. range | 35.99 | 35.43 | 33.63 | 33.50 | 33.13 | 33.21 | 33.65 |
| nugget | 0.00 | 0.00 | 0.00 | 0.00 | 0.00 | 0.00 | 0.00 |
| sill | 1.24 | 1.28 | 1.16 | 1.13 | 1.11 | 1.09 | 1.08 |
| max. lag | 60 | 60 | 60 | 60 | 60 | 60 | 60 |
| n.n. [min.,max.] | [3,20] | [3,20] | [3,20] | [3,20] | [3,20] | [3,20] | [3,20] |
| HER n.n. | 12 | 12 | 12 | 12 | 12 | 12 | 12 |
| class width | 2 | 2 | 2 | 2 | 2 | 2 | 2 |
| bin widths ($z$, $\Delta z$) | 0.2 | 0.2 | 0.2 | 0.2 | 0.2 | 0.2 | 0.2 |
| model range | 36.00 | 24.00 | 26.00 | 26.00 | 26.00 | 26.00 | 26.00 |
| $\alpha$ | 1.00 | 1.00 | 1.00 | 1.00 | 1.00 | 1.00 | 1.00 |
| $\beta$ | 0.70 | 0.60 | 0.45 | 0.40 | 0.50 | 0.65 | 0.80 |
| **Method** **Parameter**[+] | | | | SR1 | | | |
| NN n.n. | 1 | 1 | 1 | 1 | 1 | 1 | 1 |
| IDS exp. | 2 | 2 | 2 | 2 | 2 | 2 | 2 |
| OK n.n. | 12 | 12 | 12 | 12 | 12 | 12 | 12 |
| lag width | 2 | 2 | 2 | 2 | 2 | 2 | 2 |
| variogram | Spherical | Spherical | Spherical | Spherical | Spherical | Spherical | Spherical |
| eff. range | 43.53 | 35.81 | 35.43 | 34.69 | 32.70 | 32.18 | 33.30 |
| nugget | 0.28 | 0.15 | 0.18 | 0.18 | 0.17 | 0.17 | 0.20 |
| sill | 1.29 | 1.39 | 1.25 | 1.22 | 1.19 | 1.16 | 1.12 |
| max. lag | 60 | 60 | 60 | 60 | 60 | 60 | 60 |
| n.n. [min.,max.] | [3,20] | [3,20] | [3,20] | [3,20] | [3,20] | [3,20] | [3,20] |
| HER n.n. | 12 | 12 | 12 | 12 | 12 | 12 | 12 |
| class width | 2 | 2 | 2 | 2 | 2 | 2 | 2 |
| bin widths ($z$, $\Delta z$) | 0.2 | 0.2 | 0.2 | 0.2 | 0.2 | 0.2 | 0.2 |
| model range | 38.00 | 26.00 | 26.00 | 26.00 | 26.00 | 26.00 | 26.00 |
| $\alpha$ | 1.00 | 1.00 | 1.00 | 1.00 | 1.00 | 1.00 | 1.00 |
| $\beta$ | 0.70 | 0.55 | 0.60 | 0.55 | 0.55 | 0.70 | 0.80 |

[+]n.n. = number of neighbors; exp. = exponent of the weighting function; eff. range = effective range; max. = maximum; min. = minimum.

**Table S2.2: Method calibration by sample size – Parameters of the models for the long-range dataset (LR0 and LR1).**

| Model sample size | | 200 | 400 | 600 | 800 | 1000 | 1500 | 2000 |
|---|---|---|---|---|---|---|---|---|
| **Method** | **Parameter[+]** | | | | **LR0** | | | |
| NN | n.n. | 1 | 1 | 1 | 1 | 1 | 1 | 1 |
| IDS | exp. | 2 | 2 | 2 | 2 | 2 | 2 | 2 |
| OK | n.n. | 12 | 12 | 12 | 12 | 12 | 12 | 12 |
| | lag width | 2 | 2 | 2 | 2 | 2 | 2 | 2 |
| | variogram | Gaussian | Gaussian | Gaussian | Gaussian | Gaussian | Gaussian | Gaussian |
| | eff. range | 67.47 | 66.93 | 69.10 | 68.23 | 69.12 | 71.82 | 73.01 |
| | nugget | 0.00 | 0.00 | 0.00 | 0.00 | 0.00 | 0.00 | 0.00 |
| | sill | 1.06 | 0.99 | 1.03 | 1.03 | 1.05 | 1.10 | 1.10 |
| | max. lag | 100 | 100 | 100 | 100 | 100 | 100 | 100 |
| | n.n. [min.,max.] | [3,20] | [3,20] | [3,20] | [3,20] | [3,20] | [3,20] | [3,20] |
| HER[2] | n.n. | 12 | 12 | 12 | 12 | 12 | 12 | 12 |
| | class width | 2 | 2 | 2 | 2 | 2 | 2 | 2 |
| | bin widths $(z, \Delta z)$ | 0.2 | 0.2 | 0.2 | 0.2 | 0.2 | 0.2 | 0.2 |
| | model range | 46.00 | 48.00 | 48.00 | 46.00 | 46.00 | 48.00 | 48.00 |
| | $\alpha$ | 1.00 | 1.00 | 1.00 | 1.00 | 1.00 | 1.00 | 1.00 |
| | $\beta$ | 0.70 | 0.20 | 0.25 | 0.40 | 0.55 | 0.55 | 0.55 |
| **Method** | **Parameter[+]** | | | | **LR1** | | | |
| NN | n.n. | 1 | 1 | 1 | 1 | 1 | 1 | 1 |
| IDS | exp. | 2 | 2 | 2 | 2 | 2 | 2 | 2 |
| OK | n.n. | 12 | 12 | 12 | 12 | 12 | 12 | 12 |
| | lag width | 2 | 2 | 2 | 2 | 2 | 2 | 2 |
| | variogram | Gaussian | Gaussian | Gaussian | Gaussian | Gaussian | Gaussian | Gaussian |
| | eff. range | 81.79 | 76.14 | 71.43 | 69.02 | 74.43 | 78.75 | 78.05 |
| | nugget | 0.29 | 0.31 | 0.29 | 0.28 | 0.30 | 0.29 | 0.29 |
| | sill | 0.99 | 0.95 | 0.98 | 1.00 | 1.03 | 1.10 | 1.08 |
| | max. lag | 100 | 100 | 100 | 100 | 100 | 100 | 100 |
| | n.n. [min.,max.] | [3,20] | [3,20] | [3,20] | [3,20] | [3,20] | [3,20] | [3,20] |
| HER | n.n. | 12 | 12 | 12 | 12 | 12 | 12 | 12 |
| | class width | 2 | 2 | 2 | 2 | 2 | 2 | 2 |
| | bin widths $(z, \Delta z)$ | 0.2 | 0.2 | 0.2 | 0.2 | 0.2 | 0.2 | 0.2 |
| | model range | 48.00 | 46.00 | 44.00 | 44.00 | 44.00 | 46.00 | 46.00 |
| | $\alpha$ | 1.00 | 1.00 | 1.00 | 1.00 | 1.00 | 1.00 | 1.00 |
| | $\beta$ | 0.70 | 0.65 | 0.95 | 0.75 | 0.90 | 0.95 | 1.00 |

[+]n.n. = number of neighbors; exp. = exponent of the weighting function; eff. range = effective range; max. = maximum; min. = minimum.

**Supplement S3: Summary statistics of the model predictions**

This supplement summarizes the statistics of the deterministic predictions (mean of $z$) for the test set by method and learning sets (from 200 to 2000 observations). HER outcomes refer to the AND/OR aggregation. The four random fields types are presented from Table S3.1 to Table S3.4. Finally, Fig. S3.1 illustrates their residue correlation (obtained by calculating the Pearson correlation coefficient between the true values and the residue of the predictions).

**Table S3.1: Summary statistics of the prediction on test set by model – Short-range dataset without noise (SR0).**

| Method | Statistics[+] | 200 | 400 | 600 | 800 | 1000 | 1500 | 2000 |
|--------|-----------|-----|-----|-----|-----|------|------|------|
| | | | | | SR0 | | | |
| NN | mean | -0.54 | -0.55 | -0.56 | -0.56 | -0.56 | -0.56 | -0.56 |
| | sd. | 1.01 | 1.03 | 1.01 | 1.00 | 1.00 | 1.01 | 1.00 |
| | $H$ | 4.17 | 4.33 | 4.31 | 4.31 | 4.31 | 4.34 | 4.33 |
| | max. | 1.76 | 1.92 | 1.92 | 1.91 | 1.91 | 1.91 | 1.91 |
| | median | -0.44 | -0.47 | -0.57 | -0.57 | -0.53 | -0.53 | -0.52 |
| | min. | -3.68 | -3.68 | -3.68 | -3.68 | -3.68 | -3.68 | -3.68 |
| | kur. | 3.37 | 3.13 | 3.06 | 3.04 | 3.07 | 3.08 | 3.08 |
| | sk. | -0.56 | -0.43 | -0.36 | -0.30 | -0.32 | -0.30 | -0.32 |
| IDS | mean | -0.54 | -0.57 | -0.58 | -0.59 | -0.57 | -0.57 | -0.57 |
| | sd. | 0.79 | 0.88 | 0.89 | 0.90 | 0.91 | 0.93 | 0.94 |
| | $H$ | 3.96 | 4.13 | 4.16 | 4.19 | 4.21 | 4.24 | 4.26 |
| | max. | 1.58 | 1.80 | 1.79 | 1.80 | 1.80 | 1.79 | 1.80 |
| | median | -0.55 | -0.53 | -0.53 | -0.56 | -0.53 | -0.54 | -0.53 |
| | min. | -3.49 | -3.49 | -3.51 | -3.53 | -3.54 | -3.56 | -3.58 |
| | kur. | 3.56 | 3.28 | 3.27 | 3.17 | 3.15 | 3.13 | 3.10 |
| | sk. | -0.44 | -0.37 | -0.37 | -0.32 | -0.32 | -0.30 | -0.30 |
| OK | mean | -0.53 | -0.56 | -0.56 | -0.57 | -0.56 | -0.56 | -0.56 |
| | sd. | 0.86 | 0.92 | 0.93 | 0.94 | 0.95 | 0.97 | 0.97 |
| | $H$ | 4.11 | 4.21 | 4.24 | 4.26 | 4.27 | 4.30 | 4.30 |
| | max. | 1.63 | 1.86 | 1.90 | 1.90 | 1.90 | 1.90 | 1.90 |
| | median | -0.47 | -0.49 | -0.49 | -0.52 | -0.51 | -0.51 | -0.51 |
| | min. | -3.60 | -3.56 | -3.57 | -3.63 | -3.66 | -3.67 | -3.67 |
| | kur. | 3.46 | 3.18 | 3.13 | 3.09 | 3.08 | 3.08 | 3.08 |
| | sk. | -0.46 | -0.41 | -0.39 | -0.34 | -0.35 | -0.32 | -0.33 |
| HER | mean | -0.54 | -0.56 | -0.58 | -0.57 | -0.57 | -0.57 | -0.57 |
| | sd. | 0.87 | 0.95 | 0.92 | 0.96 | 0.94 | 0.98 | 0.98 |
| | $H$ | 4.08 | 4.23 | 4.21 | 4.26 | 4.24 | 4.31 | 4.31 |
| | max. | 1.70 | 1.82 | 1.81 | 1.83 | 1.82 | 1.83 | 1.86 |
| | median | -0.50 | -0.51 | -0.54 | -0.57 | -0.54 | -0.53 | -0.53 |
| | min. | -3.55 | -3.55 | -3.57 | -3.61 | -3.58 | -3.59 | -3.61 |
| | kur. | 3.54 | 3.18 | 3.22 | 3.10 | 3.13 | 3.10 | 3.07 |
| | sk. | -0.54 | -0.43 | -0.37 | -0.31 | -0.32 | -0.30 | -0.31 |

[+]sd. = standard deviation; $H$ = entropy; max. = maximum; min. = minimum; kur. = kurtosis; sk. = skewness.

**Table S3.2: Summary statistics of the prediction on test set by model – Short-range dataset with noise (SR1).**

| Method | Statistics[‡] | 200 | 400 | 600 | 800 | 1000 | 1500 | 2000 |
|--------|------------|------|------|------|------|------|------|------|
| | | | | | **SR1** | | | |
| NN | mean | -0.50 | -0.52 | -0.55 | -0.55 | -0.56 | -0.55 | -0.56 |
| | sd. | 1.15 | 1.16 | 1.14 | 1.14 | 1.13 | 1.11 | 1.11 |
| | $H$ | 4.45 | 4.51 | 4.49 | 4.50 | 4.50 | 4.48 | 4.49 |
| | max. | 2.50 | 2.70 | 2.70 | 2.70 | 2.70 | 2.70 | 2.99 |
| | median | -0.43 | -0.51 | -0.53 | -0.54 | -0.54 | -0.53 | -0.54 |
| | min. | -3.66 | -3.66 | -3.66 | -3.84 | -3.84 | -3.84 | -4.00 |
| | kur. | 2.86 | 2.79 | 2.92 | 2.91 | 2.90 | 2.97 | 2.97 |
| | sk. | -0.27 | -0.05 | -0.05 | -0.09 | -0.14 | -0.13 | -0.18 |
| IDS | mean | -0.49 | -0.53 | -0.55 | -0.58 | -0.56 | -0.56 | -0.56 |
| | sd. | 0.85 | 0.92 | 0.92 | 0.95 | 0.95 | 0.96 | 0.96 |
| | $H$ | 4.09 | 4.22 | 4.24 | 4.28 | 4.27 | 4.29 | 4.30 |
| | max. | 2.19 | 2.37 | 2.34 | 2.28 | 2.27 | 2.19 | 2.07 |
| | median | -0.47 | -0.47 | -0.50 | -0.53 | -0.51 | -0.53 | -0.52 |
| | min. | -3.42 | -3.30 | -3.29 | -3.50 | -3.52 | -3.59 | -3.55 |
| | kur. | 3.17 | 2.84 | 2.97 | 2.86 | 2.91 | 2.98 | 2.92 |
| | sk. | -0.23 | -0.13 | -0.19 | -0.21 | -0.21 | -0.22 | -0.23 |
| OK | mean | -0.49 | -0.52 | -0.54 | -0.57 | -0.55 | -0.56 | -0.56 |
| | sd. | 0.79 | 0.90 | 0.91 | 0.93 | 0.93 | 0.94 | 0.94 |
| | $H$ | 3.99 | 4.20 | 4.21 | 4.24 | 4.25 | 4.25 | 4.25 |
| | max. | 1.58 | 2.30 | 2.22 | 2.20 | 2.21 | 2.17 | 1.90 |
| | median | -0.48 | -0.46 | -0.48 | -0.51 | -0.49 | -0.49 | -0.49 |
| | min. | -3.17 | -3.16 | -3.19 | -3.31 | -3.44 | -3.51 | -3.45 |
| | kur. | 3.22 | 2.82 | 2.84 | 2.76 | 2.85 | 2.94 | 2.89 |
| | sk. | -0.22 | -0.19 | -0.24 | -0.25 | -0.26 | -0.27 | -0.26 |
| HER | mean | -0.50 | -0.53 | -0.54 | -0.57 | -0.55 | -0.56 | -0.56 |
| | sd. | 0.90 | 0.96 | 0.98 | 0.98 | 0.97 | 0.97 | 0.97 |
| | $H$ | 4.16 | 4.28 | 4.31 | 4.33 | 4.31 | 4.31 | 4.30 |
| | max. | 2.24 | 2.31 | 2.35 | 2.28 | 2.28 | 2.26 | 2.00 |
| | median | -0.47 | -0.48 | -0.50 | -0.54 | -0.51 | -0.53 | -0.52 |
| | min. | -3.32 | -3.32 | -3.38 | -3.46 | -3.45 | -3.55 | -3.54 |
| | kur. | 3.11 | 2.70 | 2.89 | 2.82 | 2.85 | 2.98 | 2.89 |
| | sk. | -0.27 | -0.13 | -0.14 | -0.16 | -0.20 | -0.19 | -0.24 |

[‡]sd. = standard deviation; $H$ = entropy; max. = maximum; min. = minimum; kur. = kurtosis; sk. = skewness.

**Table S3.3: Summary statistics of the prediction on test set by model – Long-range dataset without noise (LR0).**

| Method | Statistics[‡] | 200 | 400 | 600 | 800 | 1000 | 1500 | 2000 |
|--------|------------|------|------|------|------|------|------|------|
| | | | | | LR0 | | | |
| NN | mean | -1.03 | -1.02 | -1.01 | -1.02 | -1.02 | -1.01 | -1.02 |
| | sd. | 0.91 | 0.91 | 0.91 | 0.91 | 0.91 | 0.91 | 0.91 |
| | $H$ | 3.98 | 4.06 | 4.10 | 4.11 | 4.11 | 4.12 | 4.11 |
| | max. | 1.04 | 1.15 | 1.15 | 1.23 | 1.23 | 1.23 | 1.23 |
| | median | -0.92 | -0.91 | -0.90 | -0.90 | -0.90 | -0.90 | -0.90 |
| | min. | -2.78 | -2.78 | -3.07 | -3.07 | -3.07 | -3.08 | -3.08 |
| | kur. | 2.10 | 2.13 | 2.20 | 2.18 | 2.20 | 2.15 | 2.16 |
| | sk. | 0.00 | 0.02 | 0.03 | 0.02 | 0.03 | 0.01 | 0.00 |
| IDS | mean | -1.04 | -1.02 | -1.02 | -1.02 | -1.02 | -1.02 | -1.02 |
| | sd. | 0.85 | 0.87 | 0.88 | 0.89 | 0.89 | 0.90 | 0.90 |
| | $H$ | 3.91 | 3.98 | 4.05 | 4.07 | 4.07 | 4.08 | 4.09 |
| | max. | 0.99 | 1.08 | 1.14 | 1.15 | 1.16 | 1.14 | 1.14 |
| | median | -0.86 | -0.88 | -0.89 | -0.88 | -0.88 | -0.88 | -0.89 |
| | min. | -2.72 | -2.71 | -3.01 | -3.01 | -3.01 | -3.02 | -3.02 |
| | kur. | 1.95 | 2.01 | 2.11 | 2.12 | 2.12 | 2.11 | 2.13 |
| | sk. | -0.12 | -0.03 | -0.03 | -0.01 | -0.01 | -0.02 | -0.01 |
| OK | mean | -1.04 | -1.02 | -1.02 | -1.02 | -1.02 | -1.02 | -1.02 |
| | sd. | 0.91 | 0.91 | 0.91 | 0.91 | 0.91 | 0.91 | 0.91 |
| | $H$ | 4.11 | 4.11 | 4.12 | 4.12 | 4.12 | 4.12 | 4.12 |
| | max. | 1.34 | 1.28 | 1.24 | 1.28 | 1.27 | 1.27 | 1.27 |
| | median | -0.93 | -0.88 | -0.89 | -0.89 | -0.89 | -0.89 | -0.89 |
| | min. | -2.89 | -2.97 | -3.08 | -3.08 | -3.07 | -3.07 | -3.07 |
| | kur. | 2.12 | 2.15 | 2.17 | 2.17 | 2.16 | 2.16 | 2.16 |
| | sk. | 0.01 | 0.01 | 0.01 | 0.01 | 0.01 | 0.00 | 0.00 |
| HER | mean | -1.04 | -1.02 | -1.02 | -1.02 | -1.02 | -1.02 | -1.02 |
| | sd. | 0.88 | 0.88 | 0.89 | 0.90 | 0.90 | 0.90 | 0.91 |
| | $H$ | 3.98 | 4.03 | 4.07 | 4.09 | 4.09 | 4.09 | 4.09 |
| | max. | 1.02 | 1.13 | 1.14 | 1.22 | 1.20 | 1.15 | 1.15 |
| | median | -0.89 | -0.90 | -0.90 | -0.90 | -0.90 | -0.90 | -0.90 |
| | min. | -2.77 | -2.78 | -3.06 | -3.07 | -3.07 | -3.08 | -3.07 |
| | kur. | 2.02 | 2.09 | 2.17 | 2.16 | 2.16 | 2.13 | 2.14 |
| | sk. | -0.05 | 0.00 | 0.00 | 0.00 | 0.01 | -0.01 | -0.01 |

[‡]sd. = standard deviation; $H$ = entropy; max. = maximum; min. = minimum; kur. = kurtosis; sk. = skewness.

**Table S3.4: Summary statistics of the prediction on test set by model – Long-range dataset with noise (LR1).**

| Method | Statistics[‡] | 200 | 400 | 600 | 800 | 1000 | 1500 | 2000 |
|--------|------------|------|------|------|------|------|------|------|
| | | | | | **LR1** | | | |
| NN | mean | -1.00 | -0.99 | -1.00 | -1.01 | -1.01 | -1.00 | -1.01 |
| | sd. | 1.00 | 1.02 | 1.03 | 1.03 | 1.04 | 1.05 | 1.05 |
| | $H$ | 4.23 | 4.33 | 4.36 | 4.35 | 4.39 | 4.40 | 4.40 |
| | max. | 1.40 | 1.87 | 1.87 | 1.87 | 1.87 | 1.87 | 1.87 |
| | median | -0.90 | -0.94 | -0.97 | -0.99 | -0.99 | -0.99 | -0.98 |
| | min. | -3.19 | -3.65 | -3.65 | -3.65 | -3.65 | -3.65 | -3.87 |
| | kur. | 2.50 | 2.66 | 2.56 | 2.57 | 2.57 | 2.51 | 2.49 |
| | sk. | -0.11 | 0.03 | 0.02 | 0.10 | 0.08 | 0.06 | 0.03 |
| IDS | mean | -0.99 | -0.98 | -0.99 | -1.01 | -1.00 | -1.01 | -1.01 |
| | sd. | 0.86 | 0.90 | 0.91 | 0.92 | 0.92 | 0.93 | 0.93 |
| | $H$ | 4.04 | 4.14 | 4.14 | 4.16 | 4.18 | 4.17 | 4.16 |
| | max. | 1.21 | 1.76 | 1.48 | 1.45 | 1.61 | 1.54 | 1.43 |
| | median | -0.79 | -0.85 | -0.85 | -0.88 | -0.90 | -0.88 | -0.90 |
| | min. | -3.04 | -3.12 | -3.12 | -3.12 | -3.05 | -3.15 | -3.25 |
| | kur. | 2.21 | 2.39 | 2.28 | 2.31 | 2.32 | 2.26 | 2.26 |
| | sk. | -0.26 | 0.01 | 0.04 | 0.06 | 0.05 | 0.05 | 0.03 |
| OK | mean | -0.98 | -0.96 | -0.98 | -1.00 | -1.00 | -1.01 | -1.01 |
| | sd. | 0.79 | 0.83 | 0.85 | 0.86 | 0.87 | 0.88 | 0.89 |
| | $H$ | 3.89 | 4.01 | 4.00 | 4.02 | 4.02 | 4.04 | 4.05 |
| | max. | 0.81 | 1.29 | 1.25 | 1.32 | 1.30 | 1.14 | 1.19 |
| | median | -0.78 | -0.81 | -0.81 | -0.84 | -0.84 | -0.86 | -0.88 |
| | min. | -2.85 | -2.82 | -2.74 | -2.76 | -2.69 | -2.84 | -2.92 |
| | kur. | 2.28 | 2.38 | 2.17 | 2.18 | 2.18 | 2.13 | 2.13 |
| | sk. | -0.40 | -0.10 | -0.04 | -0.01 | -0.01 | -0.01 | -0.01 |
| HER | mean | -0.99 | -0.97 | -0.98 | -1.01 | -1.00 | -1.01 | -1.01 |
| | sd. | 0.85 | 0.89 | 0.89 | 0.90 | 0.90 | 0.92 | 0.91 |
| | $H$ | 4.01 | 4.11 | 4.07 | 4.11 | 4.11 | 4.12 | 4.11 |
| | max. | 1.20 | 1.64 | 1.32 | 1.33 | 1.36 | 1.30 | 1.30 |
| | median | -0.80 | -0.83 | -0.83 | -0.86 | -0.89 | -0.89 | -0.89 |
| | min. | -3.00 | -2.98 | -2.82 | -2.90 | -2.83 | -2.98 | -3.13 |
| | kur. | 2.21 | 2.46 | 2.23 | 2.28 | 2.27 | 2.23 | 2.23 |
| | sk. | -0.28 | 0.03 | 0.02 | 0.05 | 0.04 | 0.05 | 0.02 |

[‡]sd. = standard deviation; $H$ = entropy; max. = maximum; min. = minimum; kur. = kurtosis; sk. = skewness.

Fig. S3.1 illustrates for the residue correlation of the models calculated using the test set. The more negative the residue correlation, the greater the tendency of true $z$ values being overestimated in low-valued regions of the field and underestimated in high-valued regions.

[Figure]

**Figure S3.1: Performance comparison of NN, IDS, OK and HER: a) residue correlation for SR datasets and b) residue correlation for LR datasets. Continuous line refers to datasets without noise and dashed lines to datasets with noise.**